# The H3K4 methyltransferase Setd1b is essential for hematopoietic stem and progenitor cell homeostasis in mice

Kerstin Schmidt[1], Qinyu Zhang[2], Alpaslan Tasdogan[3,4], Andreas Petzold[5], Andreas Dahl[5], Borros M Arneth[6], Robert Slany[7], Hans Jörg Fehling[3], Andrea Kranz[2]*, Adrian Francis Stewart[2]*, Konstantinos Anastassiadis[1]*

[1]Stem Cell Engineering, Biotechnology Center, Technische Universität Dresden, Dresden, Germany; [2]Genomics, Biotechnology Center, Technische Universität Dresden, Dresden, Germany; [3]Institute of Immunology, University Hospital Ulm, Ulm, Germany; [4]Department of Dermatology, University Hospital Ulm, Ulm, Germany; [5]Deep Sequencing Group, DFG - Center for Regenerative Therapies Dresden, Dresden, Germany; [6]Institute of Laboratory Medicine and Pathobiochemistry, Molecular Diagnostics, Hospital of the Universities Giessen and Marburg, Giessen, Germany; [7]Department of Genetics, Friedrich Alexander Universität Erlangen, Erlangen, Germany

**Abstract** Hematopoietic stem cells require MLL1, which is one of six Set1/Trithorax-type histone 3 lysine 4 (H3K4) methyltransferases in mammals and clinically the most important leukemia gene. Here, we add to emerging evidence that all six H3K4 methyltransferases play essential roles in the hematopoietic system by showing that conditional mutagenesis of Setd1b in adult mice provoked aberrant homeostasis of hematopoietic stem and progenitor cells (HSPCs). Using both ubiquitous and hematopoietic-specific deletion strategies, the loss of Setd1b resulted in peripheral thrombo- and lymphocytopenia, multilineage dysplasia, myeloid-biased extramedullary hematopoiesis in the spleen, and lethality. By transplantation experiments and expression profiling, we determined that Setd1b is autonomously required in the hematopoietic lineages where it regulates key lineage specification components, including *Cebpa*, *Gata1,* and *Klf1*. Altogether, these data imply that the Set1/Trithorax-type epigenetic machinery sustains different aspects of hematopoiesis and constitutes a second framework additional to the transcription factor hierarchy of hematopoietic homeostasis.
DOI: https://doi.org/10.7554/eLife.27157.001

*For correspondence:
andrea.kranz@tu-dresden.de (AK);
francis.stewart@tu-dresden.de
(AFS);
konstantinos.anastassiadis@tu-dresden.de (KA)

**Competing interests:** The authors declare that no competing interests exist.

## Introduction

Epigenetic regulation of gene expression is key to maintain homeostasis between tissue-specific stem and progenitor cells and terminally differentiated cell derivatives (*Cedar and Bergman, 2011*). The epigenetic circuitry of stem cells serves to both maintain the stem cell transcriptional program and coordinate the gene expression networks important for driving differentiation towards distinct lineages (*Spivakov and Fisher, 2007*). Perturbations of these regulatory circuits have been associated with aging as well as aging-related diseases such as cancer (*Portela and Esteller, 2010*; *Beerman and Rossi, 2015*). Lysine methylations on histone 3 (H3) are central to the epigenetic regulation of gene expression and part of an intricate network that integrates both cooperative and antagonistic modifications (*Greer and Shi, 2012*). While trimethylation (me3) of H3K27 is generally associated with silenced gene expression (*Cao et al., 2002*), H3K4me3 marks promoters of actively transcribed genes (*Santos-Rosa et al., 2002*; *Bernstein et al., 2005*). An additional layer of

complexity is added by different levels of methylation. For instance, H3K4me2 is distributed along transcribed gene bodies, while H3K4me1 has been mainly linked to enhancer regions (*Schneider et al., 2004*; *Heintzman et al., 2007*).

In mammals, H3K4 methylation is catalyzed by six Set1/Trithorax-type histone methyltransferases that are encoded by three pairs of sister genes (*Glaser et al., 2006*): *Setd1a* and *Setd1b* (orthologous to yeast *Set1* and homologous to *Drosophila dSet*) (*Roguev et al., 2001*; *Mohan et al., 2011*), *Mll1* (*Kmt2a*) and *Mll2* (*Kmt2b*) (homologous to *Drosophila Trithorax*) (*Milne et al., 2002*), and *Mll3* (*Kmt2c*) and *Mll4* (*Kmt2d*) (homologous to a fusion of *Drosophila Cara Mitad and Trithorax-related*) (*Herz et al., 2012*; *Chauhan et al., 2012*). All six enzymes share a highly conserved catalytic SET domain and are part of large multiprotein complexes with an integral 'WRAD' core consisting of Wdr5, Rbbp5, Ash2l, and Dpy30 (*Dou et al., 2006*; *Ruthenburg et al., 2007*; *Ernst and Vakoc, 2012*). However, all six reside in separate complexes that may also include a specific subunit(s) not present in the other five (*van Nuland et al., 2013*). The proposition that each of the six mammalian Set1/Trithorax proteins exerts specialized functions to sustain increasing transcriptional complexities during multi-cellular development is supported by knockout studies in the mouse, which indicate that all six are essential for development in different ways (*Yagi et al., 1998*; *Glaser et al., 2006*; *Lee et al., 2013*; *Bledau et al., 2014*). In particular, using a multipurpose allele strategy (*Testa et al., 2004*) we recently studied the roles of the paralogs *Setd1a* and *Setd1b* during development (*Bledau et al., 2014*). Setd1a was identified as the major H3K4 methyltransferase in embryonic stem cells and peri-implantation embryos shortly before gastrulation. While Setd1a-deficient embryos were unable to develop beyond E6.5, the loss of Setd1b did not affect gastrulation but soon after provoked widespread developmental disorganization, resulting in lethality between E10.5 and E11.5. The roles of the *Set1* paralogs in the adult mammal have yet to be evaluated. However, work with cancer cell lines promoted the suggestion that Setd1a is the major H3K4 methyltransferase in all mammalian adult cell types (*Wang et al., 2009*; *Shilatifard, 2012*). The merits of this proposition need to be tested by conditional mutagenesis.

The first H3K4 methyltransferase in mammals was discovered in a high-profile race to clone the translocation break point at 11q23 associated with early-onset childhood leukemia (*Li and Ernst, 2014*). The identified translocation site fused the N-terminus of MLL1 in-frame with C-terminae of various other genes (*Ziemin-van der Poel et al., 1991*; *Meyer et al., 2009*; *Meyer et al., 2018*). Supporting the leukemia evidence for a critical role of MLL1 in hematopoiesis, mouse knockout studies showed that Mll1 is required for hematopoietic stem cell (HSC) function (*Ernst et al., 2004*; *Jude et al., 2007*). In contrast, specific deletion of Setd1a in adult long-term (LT)-HSCs via tamoxifen-inducible SCL-Cre-ER[T] is compatible with adult life and has little effect on hematopoietic maintenance. However, Setd1a-deficient LT-HSCs fail to contribute to stress-induced hematopoiesis (*Arndt et al., 2018*). The function of Setd1b in adult mice and the hematopoietic system in particular remains unassigned. In humans, mutations in the *SETD1B* gene have been detected in different kinds of malignancies including oesophageal squamous cell carcinoma (*Song et al., 2014*), gastric and colorectal cancer (*Choi et al., 2014*), endometrial carcinoma (*GarciaGarcía-Sanz et al., 2017*) and polycythemia vera (*Tiziana Storlazzi et al., 2014*). Also, recent reports uncovered a novel correlation between loss of SETD1B function and a microdeletion syndrome leading to intellectual disability (*Palumbo et al., 2015*; *Labonne et al., 2016*; *Hiraide et al., 2018*).

Here, we analyzed the function of Setd1b in adult mice using conditional mutagenesis to bypass the early death of Setd1b-deficient embryos. Using *Rosa26-Cre-ERT2* for near-ubiquitous ablation of *Setd1b* expression, we realized that the primary knockout phenotype in the adult is disturbed homeostasis of hematopoietic stem and progenitor cells (HSPCs) leading to hematopoietic failure and lethality.

## Results

### Ubiquitous loss of Setd1b in the adult mouse is lethal and affects peripheral blood composition

Previously, we reported that Setd1b is essential for post-gastrulation mouse embryonic development (*Bledau et al., 2014*). The null phenotype was obtained with both homozygous targeted constitutive and Cre recombined deletion alleles thereby indicating that both alleles are nulls. In order to study

the function of Setd1b in the adult organism, we crossed Setd1b conditional mice to the *Rosa26-Cre-ERT2* (RC) line that allows induction of the conditional knockout (cKO) in almost all tissues using tamoxifen (*Seibler et al., 2003*; *Glaser et al., 2009*). Subsequent excision of exon five by Cre recombination converts the conditional allele ('*FD*' for *F*lp and *D*re recombined) into a frameshifted cKO allele ('*FDC*' for *F*lp, *D*re and *C*re recombined) with a stop codon in exon 6 (*Figure 1A*). The breeding strategy produced both heterozygous *Setd1b^{FD/+;RC/+}* and homozygous *Setd1b^{FD/FD;RC/+}* progeny (*Figure 1B*). At an average age of 12 weeks, these mice plus additional *Setd1b^{+/+;RC/+}* wild-type (WT) controls received five doses of tamoxifen via gavage according to the outlined regime (*Figure 1B*). In comparison to WT, heterozygous *Setd1b^{FDC/+;RC/+}* animals did not present an obvious phenotype, thereby validating their further use as controls (Ctrl). Complete recombination of the *FD* allele in homozygous *Setd1b^{FDC/FDC;RC/+}* cKO animals was verified by PCR in various tissues (*Figure 1—figure supplement 1A and B*). According to the human gene database GeneCards (www.genecards.org), *SETD1B* is ubiquitously expressed in adult tissues. In cKO mice, *Setd1b* mRNA was

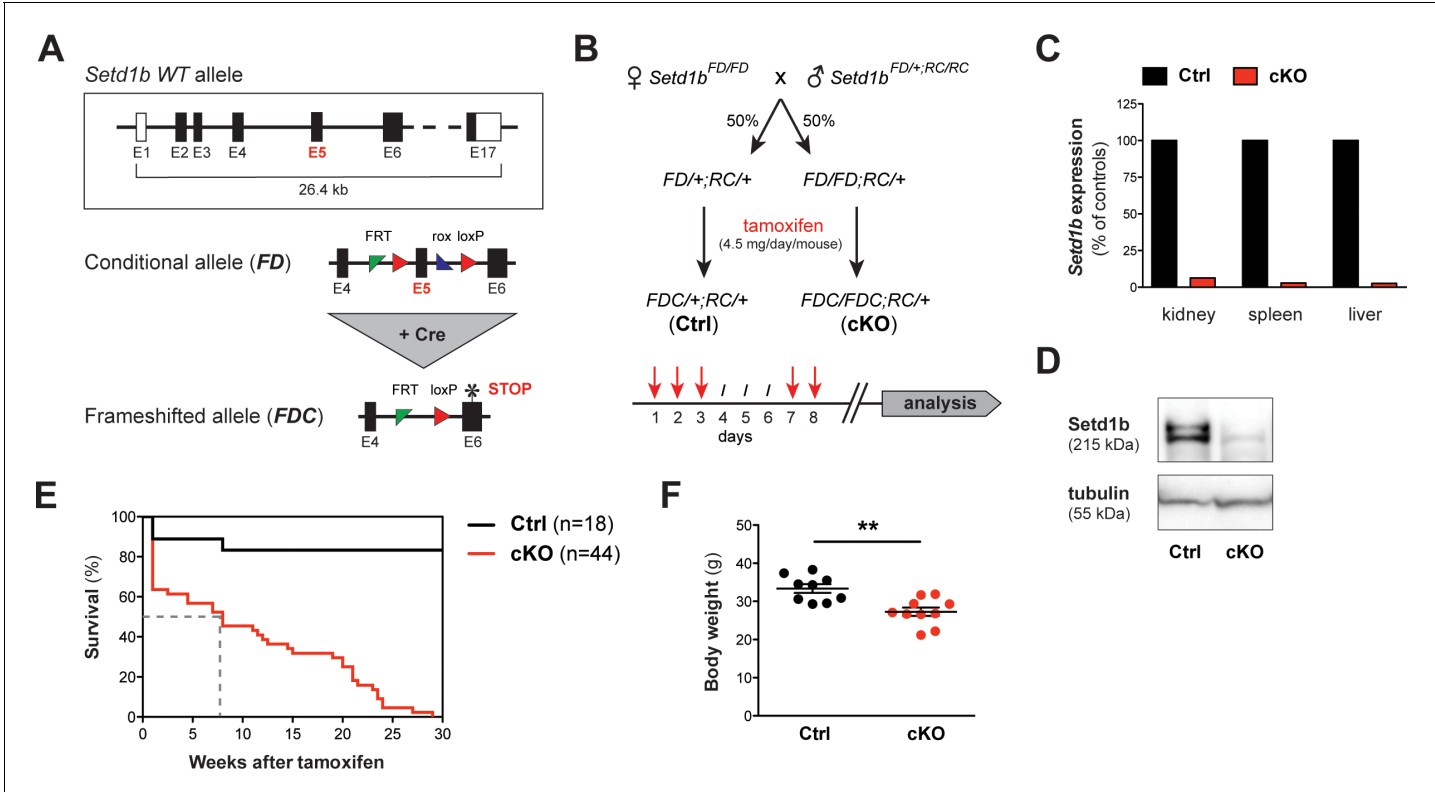

**Figure 1.** The conditional knockout (cKO) of *Setd1b* in the adult mouse is lethal. (A) *Setd1b* knockout strategy. The conditional allele (*FD*) carries loxP sites (red triangles) flanking exon 5 (E5) for Cre-mediated recombination. Excision of E5 leads to a frameshift and a premature stop codon in exon 6 (E6) (asterisk). The resulting frameshifted *FDC* allele corresponds to the null allele. Coding exons of the *Setd1b wild type* (*WT*) allele are depicted in black, non-coding exons in white. Single FRT and rox sites are the remnants of previous Flp and Dre recombination. (B) Breeding strategy and experimental setup. Female *Setd1b^{FD/FD}* mice were mated with *Setd1b^{FD/+}* males homozygous for the *Rosa26-Cre-ERT2* allele (*RC/RC*). The resulting offspring was heterozygous for *Rosa26-Cre-ERT2* (*RC/+*) and either heterozygous (*FD/+*) or homozygous (*FD/FD*) for the conditional *Setd1b* allele. To induce Cre recombination, each mouse received five doses of 4.5 mg tamoxifen via gavage over a period of 8 days according to the depicted scheme. (C) qRT-PCR analysis in kidney, spleen, and liver showed absence of *Setd1b* expression in a homozygous conditional knockout (cKO) mouse compared to a heterozygous control (Ctrl). (D) The loss of Setd1b protein (215 kDa) was validated by western blot analysis in cKO liver. (E) Kaplan-Meier survival curve. While the majority of controls (n = 18) survived, all cKO mice (n = 44) died within 30 weeks after induction with a median survival of 7–8 weeks (dashed line). (F) Male cKO mice between 12 and 26 weeks after induction (Ctrl n = 9, cKO n = 10) had significantly reduced body weights compared to control littermates (p**=0.0043). The graph depicts the mean ± SEM (data derived from five independent tamoxifen inductions).

DOI: https://doi.org/10.7554/eLife.27157.002

The following figure supplement is available for figure 1:

**Figure supplement 1.** Complete recombination is achieved in almost all tissues after tamoxifen induction.
DOI: https://doi.org/10.7554/eLife.27157.003

almost undetectable in selected organs as assessed by qRT-PCR analysis (*Figure 1C*). Furthermore, absence of Setd1b protein was confirmed by western blot (*Figure 1D*).

While most littermate controls survived, all cKO mice died within 30 weeks after tamoxifen induction (*Figure 1E*), showing progressive weakening, reduced motion and loss of body weight (*Figure 1F*). Peripheral blood counts between 7 and 27 weeks after induction revealed severe aberrations of all major blood cell lineages in cKO mice, whereas heterozygous controls remained normal. We detected a significant degree of macrocytic anemia, which was reflected by reduced numbers of red blood cells (*Figure 2A–i*), lower hemoglobin concentration (*Figure 2A–ii*), higher mean corpuscular volume (MCV) (*Figure 2A–iii*) and increased numbers of circulating reticulocytes (*Figure 2A-iv*). Also, cKO mice were highly thrombocytopenic (*Figure 2B*) and had lost more than 50% of lymphocytes (*Figure 2C–i*). In addition, neutrophil (*Figure 2C–ii*) but not eosinophil

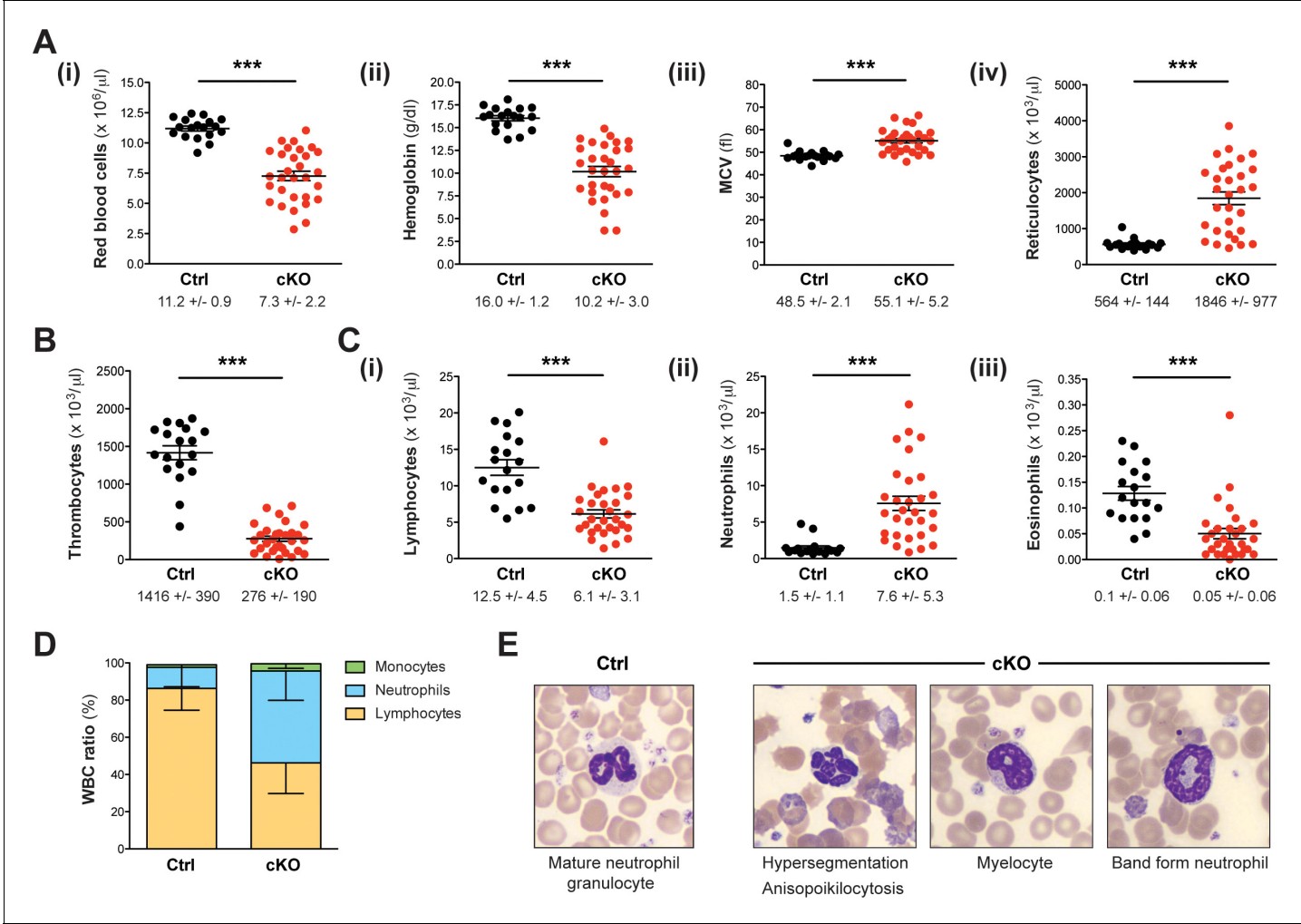

**Figure 2.** Setd1b cKO mice reveal an abnormal cellular composition in peripheral blood. (**A–C**) Peripheral blood counts of mice between 7 and 27 weeks after tamoxifen induction (Ctrl n = 18, cKO n = 30). (**A**) Red blood cell (RBC) lineage. cKO mice displayed significantly reduced RBC counts (**i**) and hemoglobin concentration (**ii**). The mean corpuscular volume (MCV) of RBCs (**iii**) as well as the number of reticulocytes (**iv**) was increased, indicating macrocytic anemia. (**B**) Thrombocyte counts were highly diminished in cKO mice. (**C**) White blood cell (WBC) lineage. While reduced numbers of lymphocytes (**i**) and eosinophil granulocytes (**iii**) were detected, neutrophil granulocytes were on average fivefold increased (**ii**) in cKO mice. Each graph depicts the mean ± SEM; the mean ± SD is indicated below (p***<0.0001). (**D**) The quantitative changes were reflected in an altered WBC ratio. The mean ± SD is illustrated. The blood count data in (**A–D**) are representative of 19 independent measurements. (**E**) Wright-Giemsa staining of blood smears. Different morphological abnormalities with respect to RBCs (anisopoikilocytosis) and neutrophil granulocytes (hypersegmented nuclei) were detected in blood from cKO mice. Furthermore, an increased presence of immature myeloid forms such as myelocytes and band form neutrophils was noted.

DOI: https://doi.org/10.7554/eLife.27157.004

granulocytes (*Figure 2C–iii*) were increased. This led to a shift of the white blood cell (WBC) ratio toward higher neutrophil granulocytes at the expense of lymphocytes (*Figure 2D*). To evaluate cellular morphologies, Wright-Giemsa staining of blood smears was performed (*Figure 2E*). Several dysplastic features in cKO samples were noticed, including altered size and shape of red blood cells (anisocytosis, poikilocytosis) as well as abnormal nuclear morphology of neutrophil granulocytes (hypersegmentation). Notably, an increased proportion of immature myeloid precursors with characteristic ring-shaped nuclei was detected. Altogether, the *Rosa26-Cre-ERT2*-mediated deletion of Setd1b in the adult mouse provoked severe defects of hematopoiesis, resulting in multilineage cytopenia - with the notable exception of neutrophil granulocytes - and dysplasia.

## Hematopoietic organs show widespread infiltration with myeloid cells of different maturation levels

Dissection of Setd1b cKO mice between 10 and 25 weeks after tamoxifen induction revealed significantly enlarged spleens but no other fatal organ abnormalities (*Figure 3A*). The normal splenic follicular structure characterized by discrete areas of red (*Figure 3B–i*) and white pulp (*Figure 3B–ii*) was obliterated. Instead, the splenic tissue was infiltrated by large cells containing irregularly shaped nuclei reminiscent of megakaryocytes (*Figure 3B–iii*) and smaller cells with ring-shaped nuclei reminiscent of granulocytes (*Figure 3B-iv*). Cellular identity was confirmed using antibodies against specific markers: von Willebrand factor (vWf) and myeloperoxidase (Mpo) for detection of megakaryocytes and granulocytes, respectively (*Figure 3C*). Of note, the widespread Mpo-staining in cKO spleens revealed varying intensities, indicating different levels of maturation. To specify the splenic subcompartments further, we performed FACS analysis using antibodies against T (CD3), B (CD19) and myeloid cells (CD11b, Gr-1) (*Figure 3D*). Consistent with the histological findings, the cKO tissue was markedly enriched for myeloid cells co-expressing CD11b and Gr-1. Most notably, both mature granulocytes (Gr-1$^{high}$) and immature precursors (Gr-1$^{low}$) were increased with respect to total spleen weight, whereas CD19+ B cells but not CD3+ T cells were decreased (*Figure 3E*). To assess *Setd1b* expression in spleen, we FACS-sorted major splenic cell populations from WT mice and performed qRT-PCR analysis using *Setd1b*-specific primers. *Setd1b* mRNA was detected in all analyzed populations with the highest expression in CD11b+ Gr-1$^{high}$ granulocytes (*Figure 3—figure supplement 1A*). In contrast to spleen, the livers of cKO mice between 10 and 25 weeks after induction did not differ in size and tissue composition from livers of age-matched controls (*Figure 3—figure supplement 1B and C*).

The average cellularity of bone marrow (BM) in cKO hind limb bones between 7 and 25 weeks after induction was comparable to that of controls (*Figure 3—figure supplement 2A*). However, cKO BM was packed with granulocytic cells, whereas control BM appeared heterogeneous due to the concomitant presence of different cell lineages (*Figure 3F*). Cytospin preparations of femoral BM further revealed an increased presence of immature stages such as myelocytes and band form neutrophils (*Figure 3G*). By immunohistochemistry, we detected an excess of Mpo+ myeloid cells as well as vWf+ megakaryocytes that appeared frayed and dysplastic in cKO BM (*Figure 3—figure supplement 2B*). The increased numbers of immature myeloid precursors (both Gr-1$^{low}$ and Gr-1$^{neg.}$) in cKO BM were confirmed by FACS analysis (*Figure 3—figure supplement 2C*). In summary, both BM and spleen were similarly affected by loss of Setd1b with widespread accumulation of myeloid cells and megakaryocytes. Of note, the BM phenotype was more pronounced at earlier stages, while the splenic contribution became more distinct at later stages, indicating a shift to compensatory extramedullary hematopoiesis.

## Hematopoietic stem and progenitor cell homeostasis is perturbed upon deletion of Setd1b

Hematopoiesis depends on the balance between hematopoietic stem and progenitor cells (HSPCs), ensuring constant and sufficient output of differentiated blood cells. Hematologic malignancies, including myeloproliferative neoplasm (MPN) and myelodysplastic syndrome (MDS), are typically characterized by disturbed HSPC homeostasis and often develop into acute myeloid leukemia (AML) (*Corey et al., 2007*; *Kitamura et al., 2014*). To evaluate whether the excess of myeloid cells in hematopoietic organs and peripheral blood of Setd1b cKO mice arises from changes in the HSPC composition, we performed FACS analysis of BM using well-characterized surface marker

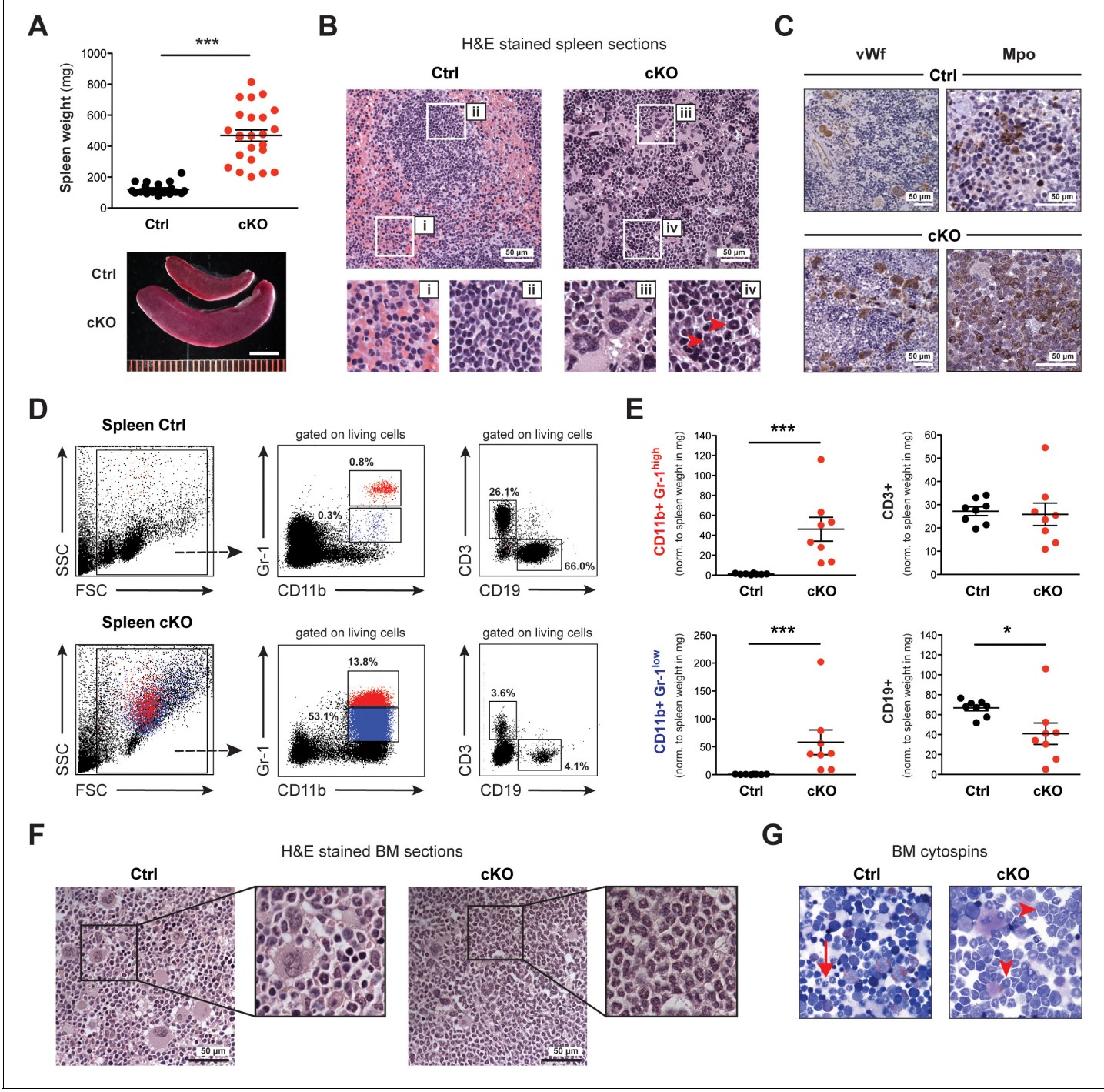

**Figure 3.** Spleen and BM of cKO mice show extensive accumulation of myeloid cells. (**A**) The spleens of cKO mice between 10 and 25 weeks after induction were on average fivefold heavier compared to age-matched controls (both n = 24) (p***<0.0001). Scale bar = 5 mm. The data is representative of 12 independent tamoxifen inductions. (**B**) H&E staining of spleen sections. Representative areas are outlined in white boxes and enlarged below: red (i) and white pulp (ii) in control spleens, megakaryocyte-like (iii) and myeloid cells (iv) with ring-shaped nuclei (red arrowheads) in cKO spleens. (**C**) Immunohistochemical staining of spleen sections with antibodies against von Willebrand factor (vWf) and myeloperoxidase (Mpo). Both vWf+ megakaryocytes and Mpo+ myeloid cells were major cell types in cKO spleens. (**D**) FACS analysis of splenic cell compartments. In cKO spleens, enriched CD11b+ Gr-1$^{high}$ (depicted in red) and CD11b+ Gr-1$^{low}$ (depicted in blue) myeloid cells as well as reduced CD3+ T cells and CD19 + B cells were detected. The given percentages are relative to living cells. (**E**) Quantification of FACS data normalized to spleen weight (mg) 7–25 weeks after induction (Ctrl n = 8, cKO n = 8). Both mature (Gr-1$^{high}$) and immature (Gr-1$^{low}$) myeloid cells were increased (p***<0.001), while B cells (p*=0.015) but not T cells were reduced in cKO tissue. The data is representative of six independent experiments. (**F**) H&E staining of femoral bone

*Figure 3 continued on next page*

*Figure 3 continued*

sections. In contrast to controls, cKO bone marrow (BM) revealed an excess of myeloid cells. (G) May-Grünwald-Giemsa staining of BM cytospin preparations. While in controls fully differentiated granulocytes were readily identified (arrow), cKO cytospins were characterized by a higher proportion of myeloid precursors (arrowheads). The graphs in (A) and (E) depict the mean ± SEM. (FSC = forward scatter, SSC = side scatter, norm. = normalized).

DOI: https://doi.org/10.7554/eLife.27157.005

The following figure supplements are available for figure 3:

**Figure supplement 1.** *Setd1b* expression in differentiated cell types and liver phenotype.

DOI: https://doi.org/10.7554/eLife.27157.006

**Figure supplement 2.** An abnormal cellular composition marks the BM of cKO mice.

DOI: https://doi.org/10.7554/eLife.27157.007

combinations (*Figure 4—figure supplement 1*). HSPCs were identified by their 'LSK' profile based on the absence of mature lineage markers (*L*in-) together with expression of *Sca-1* and *c-K*it (*Spangrude et al., 1988*). Setd1b cKO BM revealed an aberrant HSPC composition within the LSK compartment (*Figure 4A*, depicted in red). The bulk fraction consisted of CD34+ Flt3- multipotential progenitors (MPPs) (or short-term (ST)-HSCs), while CD34+ Flt3+ lymphoid primed multipotential progenitors (LMPPs) were diminished (*Figure 4B*). These relative changes coincided with an almost twofold increased LSK compartment (*Figure 4C*), which was primarily caused by an expansion of MPPs (*Figure 4D*). Furthermore, long-term (LT)-HSCs remained unchanged and LMPPs were significantly reduced in cKO hind limb bones (*Figure 4D*). In striking contrast, myeloid progenitors within the *L*in- Sca-1- c-Kit+ ('LK') compartment displayed relatively normal ratios (*Figure 4A + B*, depicted in blue). Although a slight trend toward a reduced LK compartment was observed (*Figure 4C*), no significant alterations were noted with respect to CD34+ CD16/32- CMPs (common myeloid progenitors), CD34+ CD16/32+ GMPs (granulocyte-monocyte progenitors) and CD34- CD16/32- MEPs (megakaryocyte-erythroid progenitors) (*Figure 4D*). Furthermore, we could not detect major changes in the proliferative activity of HSPCs based on BrdU incorporation 6 hr after injection (*Figure 4E*). Only LMPPs and MEPs revealed slightly increased and decreased proliferative activity, respectively. In addition, we did not observe any changes with respect to apoptosis in cKO HSPCs (data not shown). Complete recombination of the *FD* allele in selected hematopoietic cell populations was verified by PCR in FACS-sorted LT-HSCs, MPPs, LK cells, and CD11b+ Gr-1$^{high}$ granulocytes (*Figure 4—figure supplement 2A–C*). To further evaluate whether the observed homeostatic changes in the HSPC compartment correlated with different levels of *Setd1b* expression, we purified HSPCs from WT animals and performed qRT-PCR using *Setd1b*-specific primers. Interestingly, *Setd1b* was similarly expressed in all cell populations examined (*Figure 4F*). After tamoxifen induction, *Setd1b* expression was lost as shown by qRT-PCR in MPPs from cKO mice (*Figure 4—figure supplement 2D*).

Due to the discrepancy between unchanged myeloid progenitors in the LK compartment but excessive accumulation of myeloid cells in hematopoietic organs and blood of cKO mice, HSPCs in spleens were also analyzed by FACS (*Figure 4G*). Controls expectedly displayed a small population of resident LT-HSCs and MEPs. In cKO mice, however, both LSK and LK compartments were highly enriched (*Figure 4H*). This indicates that extramedullary hematopoiesis in the spleen was primarily responsible for the increased myeloid cell output in peripheral blood.

## Setd1b is intrinsically required within cells of the hematopoietic system

Because the ubiquitous *Rosa26-Cre-ERT2* mouse deletes also in cells of the hematopoietic niche, the above data do not exclude stromal contributions to the cKO phenotype. For an unbiased analysis of Setd1b function intrinsic to hematopoietic cells, we transplanted HSC-enriched (Lin- CD3-) BM of heterozygous *Setd1b$^{FD/+;RC/+}$* or homozygous *Setd1b$^{FD/FD;RC/+}$* donor mice into lethally irradiated WT (B6.SJL) recipients (*Figure 5A*). After successful reconstitution of the hematopoietic system, the first cohort was subjected to tamoxifen induction 20 weeks after transplantation (TX) and the second cohort 28 weeks after TX (*Figure 5B*). Donor and host cells were distinguished by FACS analysis of the surface markers CD45.2 and CD45.1, respectively (*Figure 5C*, cohort 1; *Figure 5—figure supplement 1*, cohort 2). After induction, the overall number of CD45.2+ donor derived cells continuously dropped in blood from cKO transplanted mice, whereas in controls repopulation with CD45.2 + cells reached >90% within a few weeks (*Figure 5C–i; Figure 5—figure supplement 1A*). Because

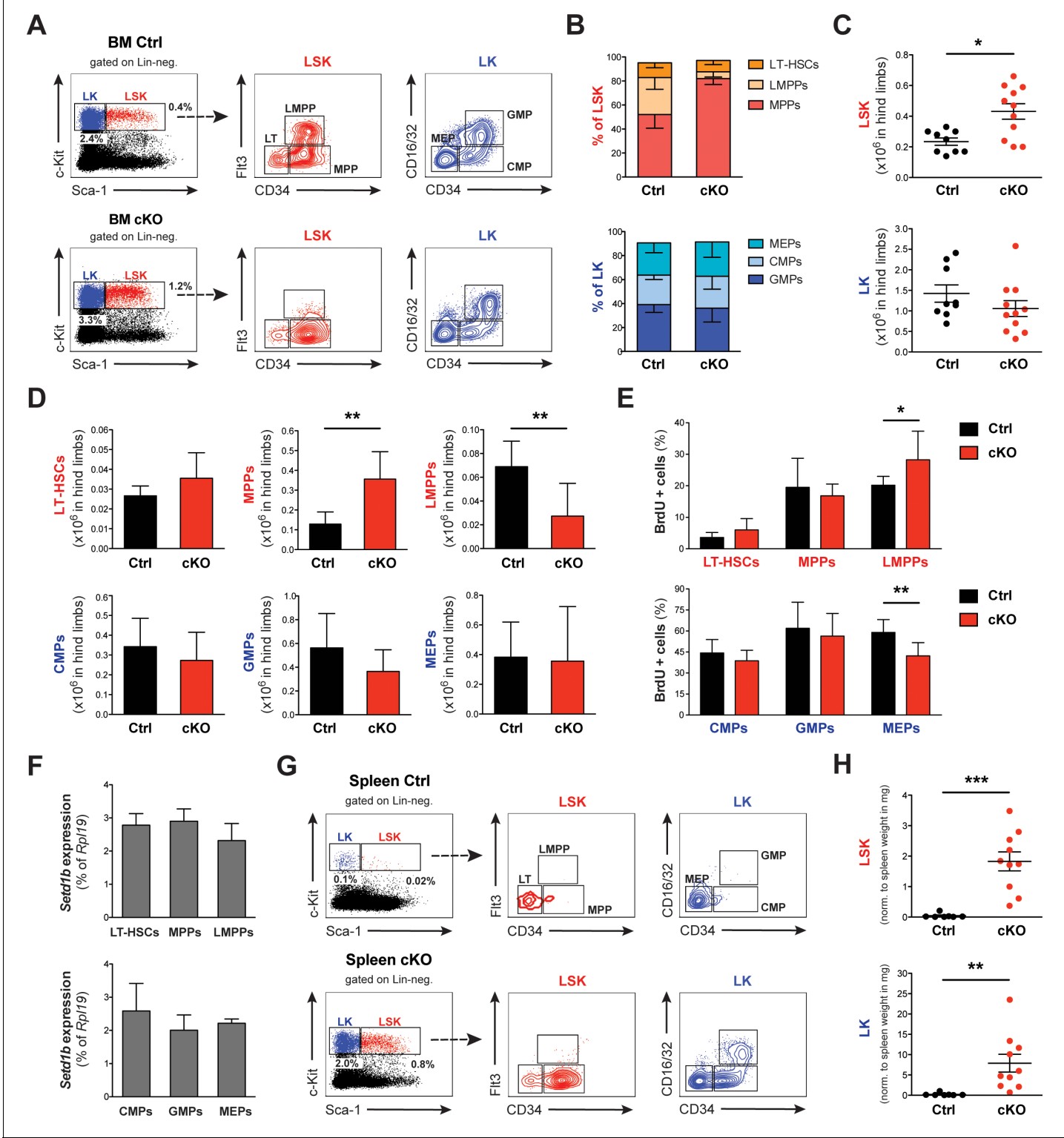

**Figure 4.** Setd1b-deficiency leads to disturbed HSPC homeostasis in BM and spleen. (**A**) FACS analysis of hematopoietic stem and progenitor cell (HSPC) populations in BM. The LSK (Lin- Sca-1+ c-Kit+) compartment (depicted in red) is subdivided into LT-HSCs, MPPs, and LMPPs based on differential expression of CD34 and Flt3. The LK (Lin- Sca-1- c-Kit+) population (depicted in blue) divides into CMPs, GMPs, and MEPs based on differential expression of CD34 and CD16/32. The given percentages are relative to total cells analyzed. (**B**) HSPC ratios 7–25 weeks after tamoxifen induction (Ctrl n = 9, cKO n = 11). The LSK ratio in cKO BM shifted toward a higher percentage of MPPs at the expense of LMPPs. (**C**) Quantification of FACS data with respect to total cellularity in both hind limb bones (x10⁶). In cKO BM, the LSK compartment has roughly doubled (p*=0.011). (**D**) This

*Figure 4 continued on next page*

*Figure 4 continued*

increase resulted from an enriched pool of MPPs (p**=0.001), while LMPPs were significantly reduced (p**=0.006). Of note, none of the myeloid progenitor populations were majorly affected. Data in (B–D) are representative of six independent experiments. (E) BrdU proliferation assay 6 hr after injection (Ctrl n = 7, cKO n = 7). Only in case of LMPPs (p**=0.038) and MEPs (p**=0.007) minor changes were detected in cKO BM. The data were derived from five independent experiments. (F) qRT-PCR analysis of *Setd1b* expression. Similar levels of *Setd1b* mRNA were detected in all FACS-sorted HSPC populations from WT mice (n = 3) relative to *Rpl19* expression. (G) FACS analysis of HSPCs in the spleen. Other than in controls, the c-Kit + compartment was highly enriched in cKO spleens, indicating fully active hematopoiesis. The given percentages are relative to total cells analyzed. (H) Quantification of FACS data normalized to spleen weight (mg) 7–25 weeks after induction (Ctrl n = 7, cKO n = 10). Both LSK (p***<0.001) and LK compartments (p**=0.001) were significantly increased in cKO spleens. The data are representative of six independent experiments. The graphs in (B) and (D)-(F) depict the mean ± SD; in (C) and (H) the mean ± SEM is provided.

DOI: https://doi.org/10.7554/eLife.27157.008

The following source data and figure supplements are available for figure 4:

**Source data 1.** Numerical data used to generate *Figure 4B,C and D* (HSPCs BM RC).
DOI: https://doi.org/10.7554/eLife.27157.011
**Figure supplement 1.** HSPC surface markers and gating strategy.
DOI: https://doi.org/10.7554/eLife.27157.009
**Figure supplement 2.** Complete recombination is achieved in FACS-sorted hematopoietic cell populations after tamoxifen induction.
DOI: https://doi.org/10.7554/eLife.27157.010

the CD45.2+ fraction of CD11b+ Gr-1[high] granulocytes was unaffected (*Figure 5C–ii*; *Figure 5—figure supplement 1B*), we assume that the progressive loss of CD45.2+ cells in blood is due to an overall decrease in lymphocyte counts (*Figure 5D–i*). Furthermore, cKO transplanted mice revealed nearly normal neutrophil (*Figure 5D–ii*) and red blood cell counts (*Figure 5D–iii*), but significantly reduced thrombocyte counts (*Figure 5D-iv*), in accordance with our findings after ubiquitous deletion of Setd1b (*Figure 2B*). Likewise, dysplastic granulocytes and immature myeloid forms were detected in blood (*Figure 5E*). Despite these obvious hematopoietic defects, the majority of cKO transplanted mice appeared normal and survived long-term; only two animals deceased (14.3%) 21 and 47 weeks after tamoxifen induction, respectively. The remaining mice were subsequently analyzed at 22 (cohort 1) and 48 weeks (cohort 2) after induction. Except for mildly increased spleen weights (*Figure 5F*), no major organ abnormalities were noted in cKO transplanted mice. However, FACS analysis of splenic cell populations largely recapitulated our previous findings, including reduced numbers of CD19+ B cells and increased numbers of CD11b+ Gr-1[high] granulocytes and CD11b+ Gr-1[low] myeloid precursors (*Figure 5G*). Similarly, the characteristic shift of HSPCs within the LSK compartment was reproduced in BM from cKO transplanted mice (*Figure 5H*). Although total numbers of LT-HSCs as well as CMPs, GMPs, and MEPs did not differ between cKO transplanted mice and controls, MPPs were markedly enriched and LMPPs nearly completely lost (*Figure 5I*). In summary, the confined deletion of Setd1b within the hematopoietic system of cKO transplanted mice was sufficient to generate major characteristics of the cKO phenotype (lympho- and thrombocytopenia, dysplasia, myeloid skewing in spleen, disturbed HSPC homeostasis). This suggests that Setd1b function is largely intrinsic to the hematopoietic lineages.

## The hematopoietic-specific deletion of Setd1b by *Vav-Cre* is lethal and reveals functional deficits of HSPCs

Until this point, we induced the knockout of Setd1b in adult mice that had already established a functional hematopoietic system. To examine whether Setd1b function is required for hematopoietic development, we crossed Setd1b conditional mice to the hematopoietic-specific *Vav-Cre* (VC) deleter, which activates Cre expression in all hematopoietic lineages from E13.5 onwards (*Ogilvy et al., 1999*; *Stadtfeld and Graf, 2005*). Both heterozygous *Setd1b[FD/+;VC/+]* (Ctrl) and homozygous *Setd1b[FD/FD;VC/+]* (vKO) littermates were born at expected Mendelian ratios (*Figure 6A*). Complete recombination of the *FD* allele (to *FDC*) in selected hematopoietic lineages was verified by PCR in FACS-sorted LT-HSCs, MPPs, LK cells, and CD11b+ Gr-1[high] granulocytes (*Figure 6—figure supplement 1*). At the time of weaning, vKO pups did not display obvious abnormalities, indicating that an adult hematopoietic system could be established in the absence of Setd1b. However, vKO mice died prematurely with a median survival of 25 weeks (*Figure 6B*). Analysis of peripheral blood counts in 10–30 weeks old mice revealed multilineage cytopenia affecting lymphocytes, thrombocytes, and

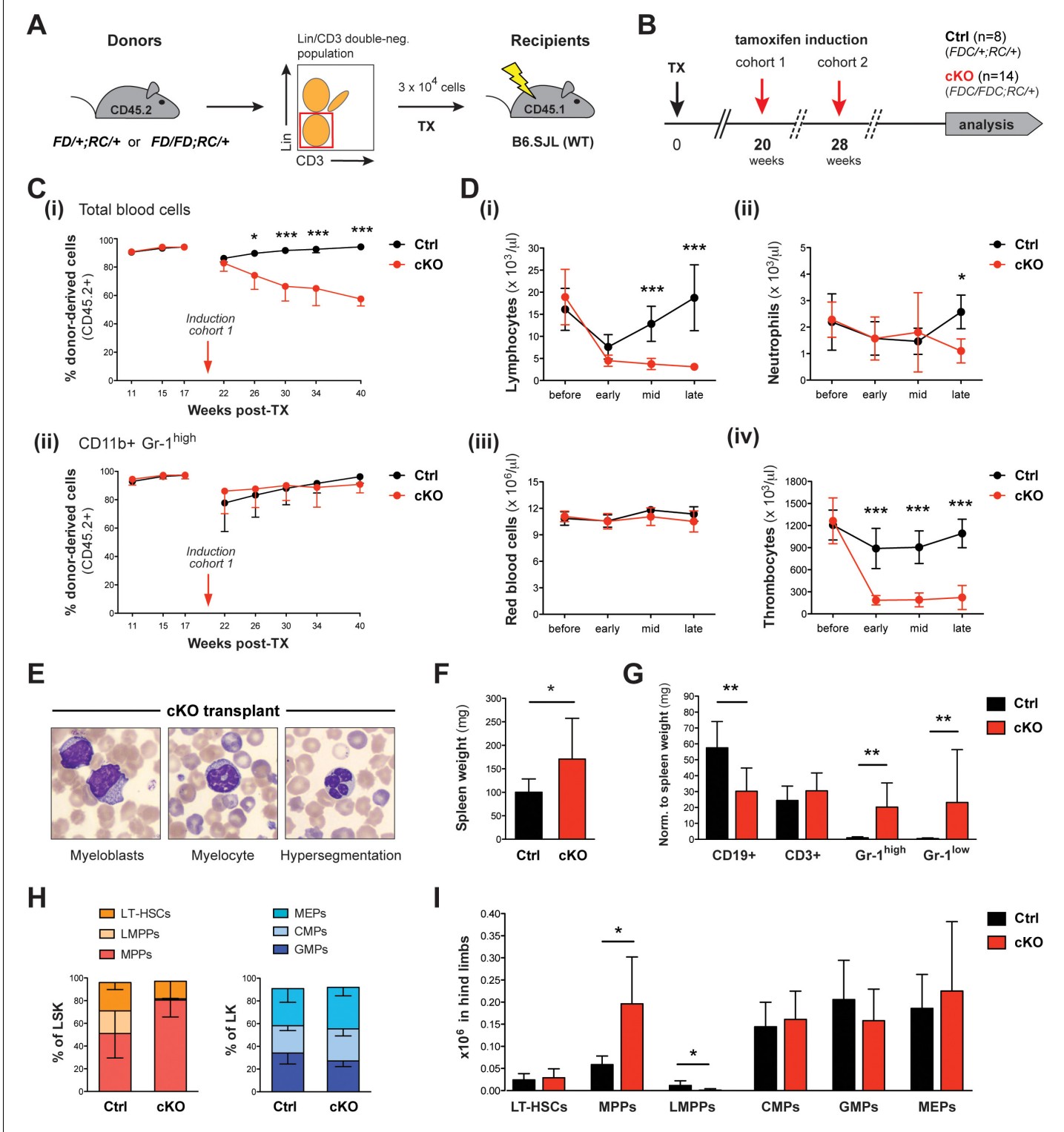

**Figure 5.** Upon tamoxifen induction, mice with a cKO BM transplant reproduce major characteristics of the cKO phenotype. (**A**) Transplantation (TX) setup. The HSC-enriched fraction (Lin- CD3-) of BM from heterozygous *Setd1b^FD/+;RC/+* or homozygous *Setd1b^FD/FD;RC/+* donor mice (CD45.2+) was intravenously injected into lethally irradiated B6.SJL recipients (CD45.1+) (3 × 10^4 cells per TX). (**B**) Successfully reconstituted mice (Ctrl n = 8, cKO n = 14) were induced with tamoxifen: cohort 1 at 20 weeks, cohort 2 at 28 weeks after TX (each cohort with Ctrl n = 4, cKO n = 7). (**C**) FACS analysis of the CD45.1/2 chimerism in peripheral blood (cohort 1). (**i**) Before induction, high percentages (92–95%) of donor-derived CD45.2+ blood cells were reached in all transplanted mice. Immediately after induction, this ratio declined in cKO transplanted mice, whereas it remained high in controls

*Figure 5 continued on next page*

*Figure 5 continued*

(p*<0.05, p***<0.001). (ii) The contribution of CD45.2+ cells to CD11b+ Gr-1$^{high}$ granulocytes was comparable between cKO transplanted mice and controls. (D) Blood count analysis, including data from both cohorts, before and after induction (early = 4–6 weeks, mid = 11–12 weeks, late = 21–22 weeks). (i) After an initial decline, cKO transplanted mice remained lymphocytopenic. (ii) Neutrophil granulocytes as well as (iii) red blood cells revealed normal levels except for a slight decrease at late stages. (iv) Platelet counts were highly diminished at all stages (p*<0.05, p***<0.001). (E) Wright-Giemsa stained blood smears of cKO transplanted mice 48 weeks after induction. Both immature forms and nuclear dysplasia of myeloid cells were noted. (F) cKO transplanted mice showed mildly increased (p*=0.044) spleen weights (Ctrl n = 8, cKO n = 12; 22 + 48 weeks after induction). (G) FACS analysis of splenic cell compartments. While CD11b+ Gr-1$^{high}$ and CD11b+ Gr-1$^{low}$ myeloid cells were increased (p**=0.003), CD19+ B cells but not CD3+ T cells were decreased (p**=0.002) in cKO transplanted mice. (H) FACS analysis of HSPC populations in the BM. cKO transplanted mice revealed abnormal HSPC ratios within the LSK compartment. (I) While MPPs were enriched (p*=0.019), LMPPs were almost undetectable (p*=0.011). All graphs depict the mean ± SD (data derived from two independent experiments).

DOI: https://doi.org/10.7554/eLife.27157.012

The following figure supplement is available for figure 5:

**Figure supplement 1.** FACS analysis of the CD45.1/2 chimerism in peripheral blood (cohort 2).

DOI: https://doi.org/10.7554/eLife.27157.013

- to a minor degree - red blood cells (*Figure 6C*). Despite myeloid precursor increase (*Figure 6D*), the overall quantity of neutrophil granulocytes in peripheral blood remained unchanged (*Figure 6C*), which is similar to our findings in cKO transplanted mice (*Figure 5D–ii*). Likewise, only mild splenomegaly was observed in ≥12 weeks old vKO mice compared to respective controls (*Figure 6E*). The splenic cell compartments, however, shifted toward increased CD11b+ Gr-1+ myeloid cells and decreased lymphocyte populations (*Figure 6F*). HSPC analysis in vKO BM revealed an almost complete loss of LMPPs along with significantly reduced GMP numbers (*Figure 6G*). In striking contrast to our findings in cKO mice, no selective expansion of the MPP compartment was detected.

To characterize the properties of Setd1b-deficient HSPCs independent from the BM niche, we performed colony-forming unit (CFU) assays in semi-solid medium that specifically supported myeloid differentiation. The number of CFUs decreased with increasing age of vKO mice, indicating progressive exhaustion of stem cells in the BM (*Figure 6—figure supplement 2A*). In serial replating experiments, c-Kit+ control cells completely exhausted their self-renewal capacity after three rounds of replating. On the other hand, vKO cells could still form colonies, indicating delayed differentiation of myeloid progenitors (*Figure 6—figure supplement 2B-i*). Instead, when challenged by a strong oncogenic input such as *MLL-ENL*, c-Kit+ vKO cells eventually exhausted their replating potential in contrast to control cells that were immortalized after four rounds of replating (*Figure 6—figure supplement 2B-ii*). Apparently, vKO myeloid progenitors were incompatible with the high proliferative demand imposed on them by *MLL-ENL*, indicating that the loss of Setd1b not only causes differentiation but also proliferation defects. This functional impairment potentially leads to a chronic stress situation *in vivo*, marked by insufficient output of differentiated blood cells and increased compensatory hematopoietic activity. The inability of vKO mice to tolerate this ongoing stress response provides a likely explanation for the finding that 14% of vKO mice older than 24 weeks developed a myeloproliferative neoplasm (MPN). The disease manifested in rapid lethality, highly increased spleen weights, leukemic liver infiltrations, and an almost exclusive presence of myeloid blasts in affected organs (*Figure 6—figure supplement 2C–E*).

## HSPCs, especially LT-HSCs and LMPPs, depend on Setd1b function

To further specify the functional defects of vKO HSPCs, we transplanted total BM cells from 12 weeks old heterozygous *Setd1b$^{FD/+;VC/+}$* (Ctrl) or homozygous *Setd1b$^{FD/FD;VC/+}$* (vKO) donor mice (CD45.2+) into lethally irradiated WT (B6.SJL) recipients (CD45.1+) (*Figure 7A*). Reconstitution of the hematopoietic system was assessed by FACS analysis of CD45.2+ donor-derived blood cells at weeks 4, 8, and 14 after TX (*Figure 7B*). While in controls the contribution of CD45.2+ blood cells reached >90%, it significantly dropped in vKO transplanted mice to <40% within 14 weeks after TX (*Figure 7B–i*). However, this decrease did not affect CD11b+ Gr-1$^{high}$ granulocytes (*Figure 7B–ii*), indicating that vKO HSPCs were generally able to repopulate the irradiated BM niche. Instead, the loss of CD45.2+ cells in blood was almost entirely attributable to lymphocyte populations, of which donor-derived CD19+ B cells were completely undetectable in vKO transplanted mice (*Figure 7C*). Similar to the findings in vKO mice (*Figure 6C*), both lymphocyte and thrombocyte counts were

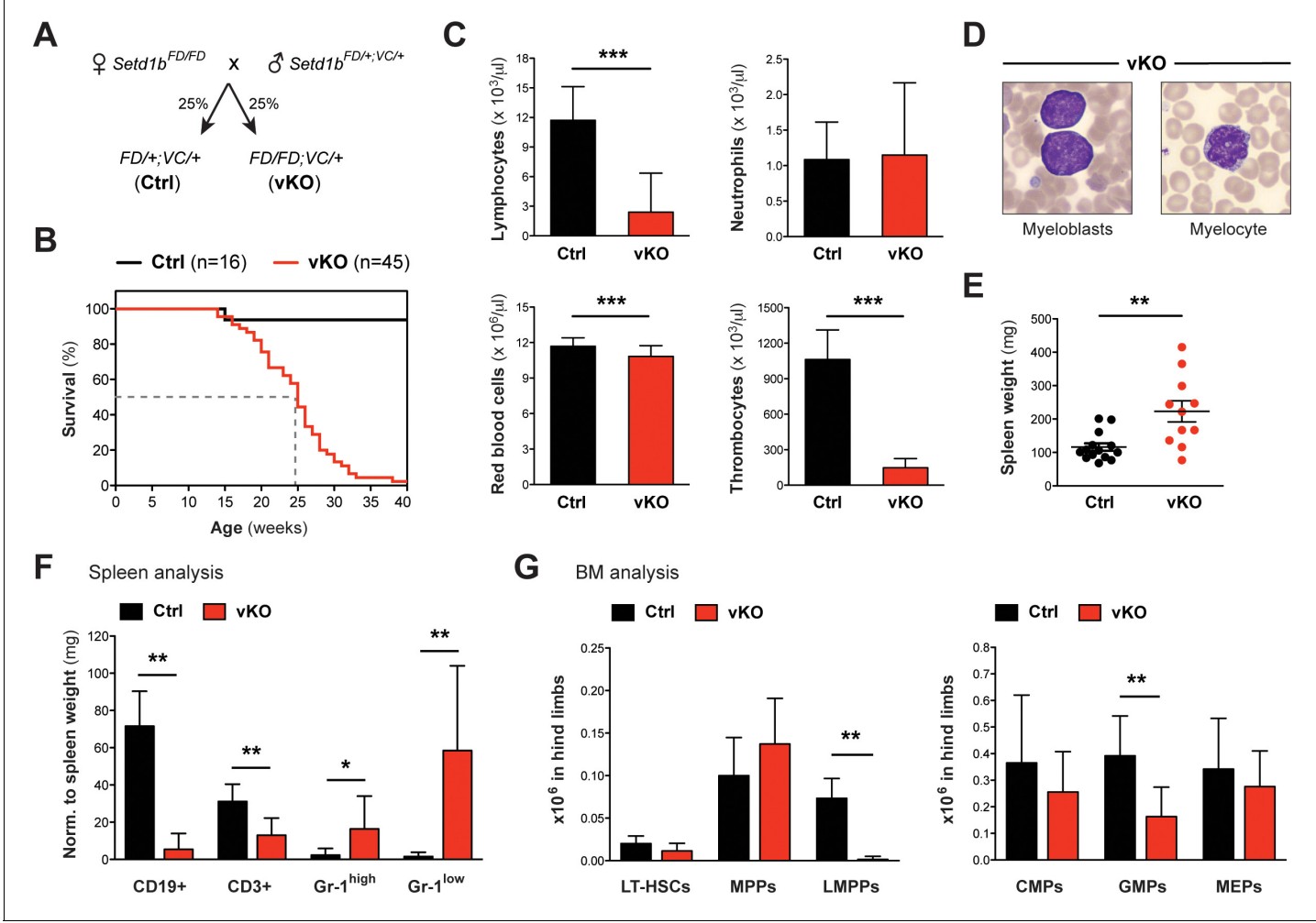

**Figure 6.** Hematopoietic-specific deletion of Setd1b using *Vav-Cre* is lethal and impedes production of early progenitors. (A) Breeding strategy. Mice heterozygous for *Vav-Cre* (*VC/+*) and either heterozygous (Ctrl) or homozygous (vKO) for the conditional *Setd1b* allele were used for analysis. (B) Kaplan-Meier survival curve. All vKO mice died with a median survival of 25 weeks (dashed line). (C) Peripheral blood counts of mice aged between 10 and 32 weeks (Ctrl n = 30, vKO n = 35). Both lymphocyte and thrombocyte counts were severely diminished in vKO blood (p***<0.0001). Furthermore, mice showed mild but significant anemia (p***=0.0002) (data representative of >20 measurements). (D) Wright-Giemsa stained blood smears. Immature myeloid forms, including myeloblasts and myelocytes, were detected in vKO blood. (E) The majority of vKO mice (n = 11) aged between 5 and 31 weeks developed mild splenomegaly as compared to controls (n = 14) (p**=0.004). (F) FACS analysis of splenic cell compartments (Ctrl n = 6, vKO n = 7). In spleens from vKO mice, lymphocytes were highly reduced with respect to spleen weight (mg) (p**<0.008), while CD11b+ Gr-1high (p*=0.035) as well as CD11b+ Gr-1low granulocytes (p**=0.001) were increased. (G) FACS analysis of HSPCs in BM. In vKO mice (n = 7), LMPPs were almost undetectable (p**=0.003) and GMPs were more than twofold decreased (p**=0.005) compared to control littermates (n = 6). The data in (F) and (G) are representative of four independent experiments. The graphs in (C), (F), and (G) depict the mean ± SD; in (E) the mean ± SEM is illustrated.

DOI: https://doi.org/10.7554/eLife.27157.014

The following source data and figure supplements are available for figure 6:

**Source data 1.** Numerical data used to generate *Figure 6G* (HSPCs BM VC).
DOI: https://doi.org/10.7554/eLife.27157.017
**Figure supplement 1.** Complete recombination is achieved in FACS-sorted hematopoietic cell populations.
DOI: https://doi.org/10.7554/eLife.27157.015
**Figure supplement 2.** Setd1b-deficient HSPCs reveal impaired differentiation and proliferation.
DOI: https://doi.org/10.7554/eLife.27157.016

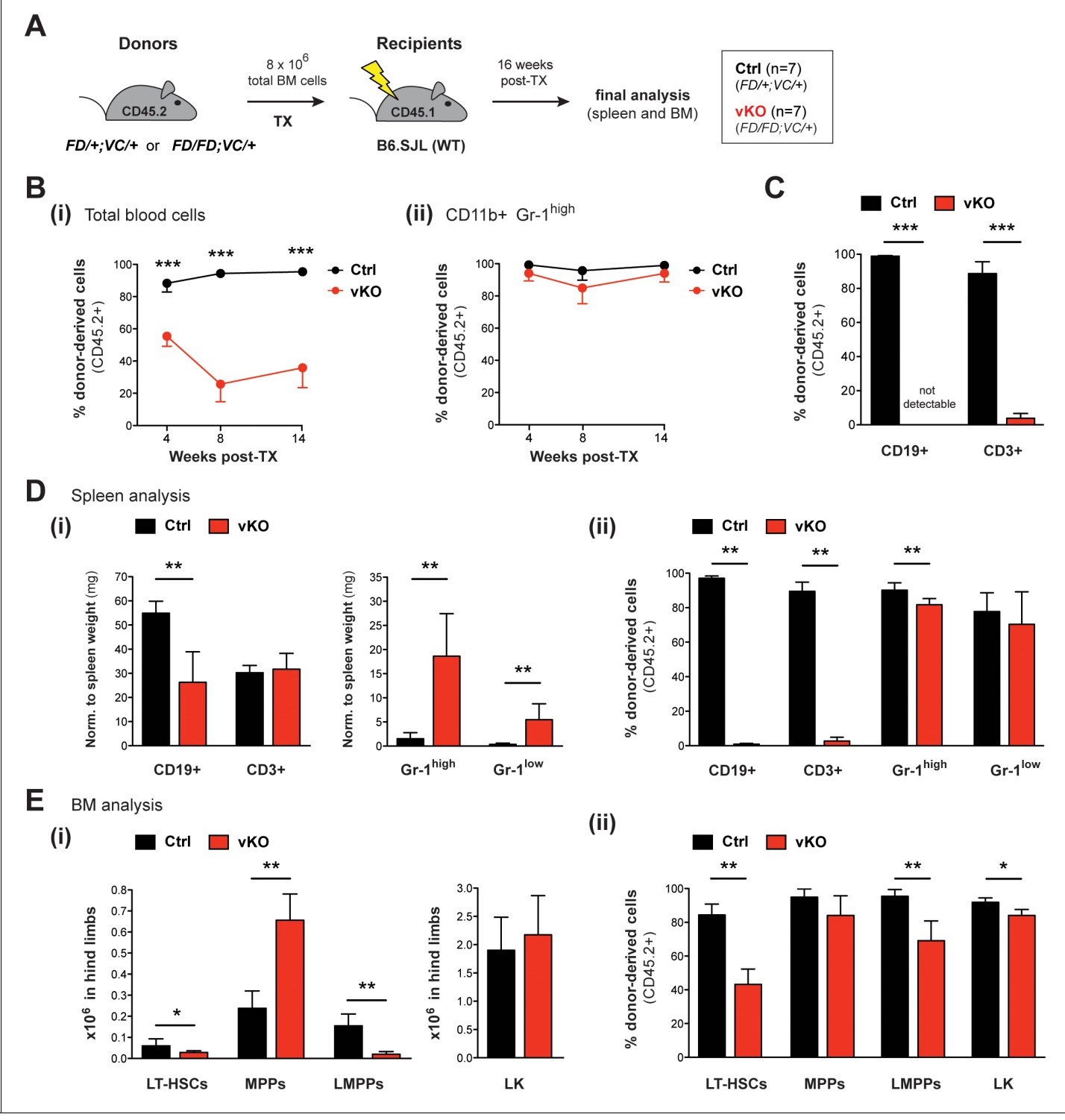

**Figure 7.** Transplantation of vKO BM recapitulates major characteristics of the vKO phenotype. (A) Transplantation (TX) setup. BM cells from heterozygous *Setd1b^FD/+;VC/+* or homozygous *Setd1b^FD/FD;VC/+* donor mice (CD45.2+) were intravenously injected into lethally irradiated B6.SJL recipients (CD45.1+) ($8 \times 10^6$ cells per TX). The mice (Ctrl n = 7, vKO n = 7) were analyzed 16 weeks after TX. (B) FACS analysis of the CD45.1/2 chimerism in peripheral blood over time. (i) The contribution of CD45.2+ donor derived cells severely declined in vKO transplanted mice (p***<0.001). (ii) In CD11b+ Gr-1^high granulocytes, the CD45.2+ ratio remained high and was comparable between vKO transplanted mice and controls. (C) FACS analysis of the CD45.1/2 chimerism in peripheral blood lymphocytes 14 weeks after TX. Both CD19+ B cell and CD3+ T cell populations were almost completely devoid of CD45.2+ donor-derived cells in vKO transplanted mice (p***=0.001). (D) FACS analysis of splenic cell compartments (Ctrl n = 6,
*Figure 7 continued on next page*

*Figure 7 continued*

vKO n = 5) 16 weeks after TX. (**i**) In spleens from vKO transplanted mice, B cells were significantly reduced (p**=0.004) with respect to spleen weight (mg), while both CD11b+ Gr-1$^{high}$ as well as CD11b+ Gr-1$^{low}$ granulocytes were increased (p**=0.004). (**ii**) FACS analysis of the CD45.1/2 chimerism in the respective cell compartments. While the contribution of CD45.2+ donor-derived cells in lymphocytes from vKO transplanted mice was almost completely abolished (p**=0.004), it was only slightly reduced in Gr-1$^{high}$ granulocytes (p**=0.004) and unchanged in Gr-1$^{low}$ cells. (**E**) FACS analysis of HSPCs in BM (Ctrl n = 6, vKO n = 5) 16 weeks after TX. (**i**) In vKO transplanted mice, significant reductions were noted in both LT-HSCs (p*=0.028) and LMPPs (p**=0.005), while MPPs were increased (p**=0.009) and LK cells were unchanged compared to controls. (**ii**) FACS analysis of the CD45.1/2 chimerism in the respective cell compartments. Except for MPPs, the contribution of CD45.2+ donor derived cells was significantly reduced in LT-HSCs (p**=0.004), LMPPs (p**=0.004), and LK cells (p*=0.017) in BM from vKO transplanted mice. All graphs depict the mean ± SD (data in (**D**) and (**E**) derived from three independent experiments).

DOI: https://doi.org/10.7554/eLife.27157.018

The following figure supplement is available for figure 7:

**Figure supplement 1.** Blood count analysis and spleen weights in *VC/+* transplanted mice.

DOI: https://doi.org/10.7554/eLife.27157.019

highly diminished, while neutrophil granulocytes as well as red blood cells were unchanged in vKO transplanted mice (*Figure 7—figure supplement 1A*). Final analysis of hematopoietic cell populations in spleen and BM was performed 16 weeks after TX. Of note, until this time, no vKO transplanted mouse died or developed MPN. While average spleen weights were normal (*Figure 7—figure supplement 1B*), vKO transplanted mice revealed decreased total numbers of CD19+ B cells but not CD3+ T cells and increased numbers of both CD11b+ Gr-1$^{high}$ and CD11b+ Gr-1$^{low}$ granulocytes in spleen (*Figure 7D–i*). Similar to blood, the contribution of CD45.2+ donor-derived cells in splenic lymphocytes was almost completely abolished, while it was barely affected in CD11b+ Gr-1$^{high}$ and CD11b+ Gr-1$^{low}$ granulocytes (*Figure 7D–ii*). Final BM analysis of vKO transplanted mice revealed decreased numbers of both LT-HSCs and LMPPs along with increased numbers of MPPs and an unchanged LK compartment (*Figure 7E–i*). Interestingly, the contribution of CD45.2+ donor-derived cells in HSPCs in vKO transplanted mice was significantly reduced in all populations examined except for MPPs (*Figure 7E–ii*). Altogether, these findings substantiate the crucial role of Setd1b in HSPC populations, including LT-HSCs and LMPPs. In particular, lymphopoiesis appears to fully depend on Setd1b function.

## Competitive BM reconstitution reveals cell-autonomous effects of Setd1b in both myeloid and lymphoid lineages

The data gathered so far largely reflect Setd1b hematopoietic contributions during adaptation to a failing hematopoietic system. All these Setd1b KO models were characterized by varying degrees of multilineage peripheral cytopenia, which likely provoked compensatory hematopoietic activity. Therefore, several aspects of the KO phenotype might be due to stress hematopoiesis and not directly relate to cell-autonomous effects of Setd1b. To differentiate between cell-autonomous and non-autonomous effects of Setd1b deletion in the absence of peripheral cytopenia, we performed competitive BM reconstitution experiments using the *Rosa26-Cre-ERT2* model. Total BM cells from heterozygous *Setd1b$^{FD/+;RC/+}$* or homozygous *Setd1b$^{FD/FD;RC/+}$* donor mice were mixed in a 1:1 ratio with WT BM and transplanted into lethally irradiated WT (B6.SJL) recipients (*Figure 8A*). After successful reconstitution of the hematopoietic system, the mice were subjected to tamoxifen induction 16 weeks after TX. The hematopoietic contribution of CD45.2+ donor-derived cells was regularly monitored by FACS analysis in peripheral blood (*Figure 8B*). Immediately after tamoxifen induction, the percentage of CD45.2+ cells in blood severely declined in cKO/WT transplanted mice (*Figure 8B–i*). In striking contrast to previous transplantations, this decline was also reflected in CD11b+ Gr-1$^{high}$ granulocytes (*Figure 8B–ii*). In addition, the contribution of CD45.2+ cells in both CD19+ B cells and CD3+ T cells was significantly reduced (*Figure 8C*). Final analysis of hematopoietic cell populations in spleen and BM was performed 14 weeks after tamoxifen induction (30 weeks after TX). In accordance with the competitive setting, cKO/WT transplanted mice showed normal numbers of lymphocytes and granulocytes in blood (data not shown) as well as in spleen (*Figure 8D–i*). However, the contribution of CD45.2+ cells in each cell population was significantly decreased (*Figure 8D–ii*). In BM of cKO/WT transplanted mice, normal numbers of LT-HSCs, LMPPs, and LK cells were detected except for an enriched MPP population (*Figure 8E–i*). Interestingly,

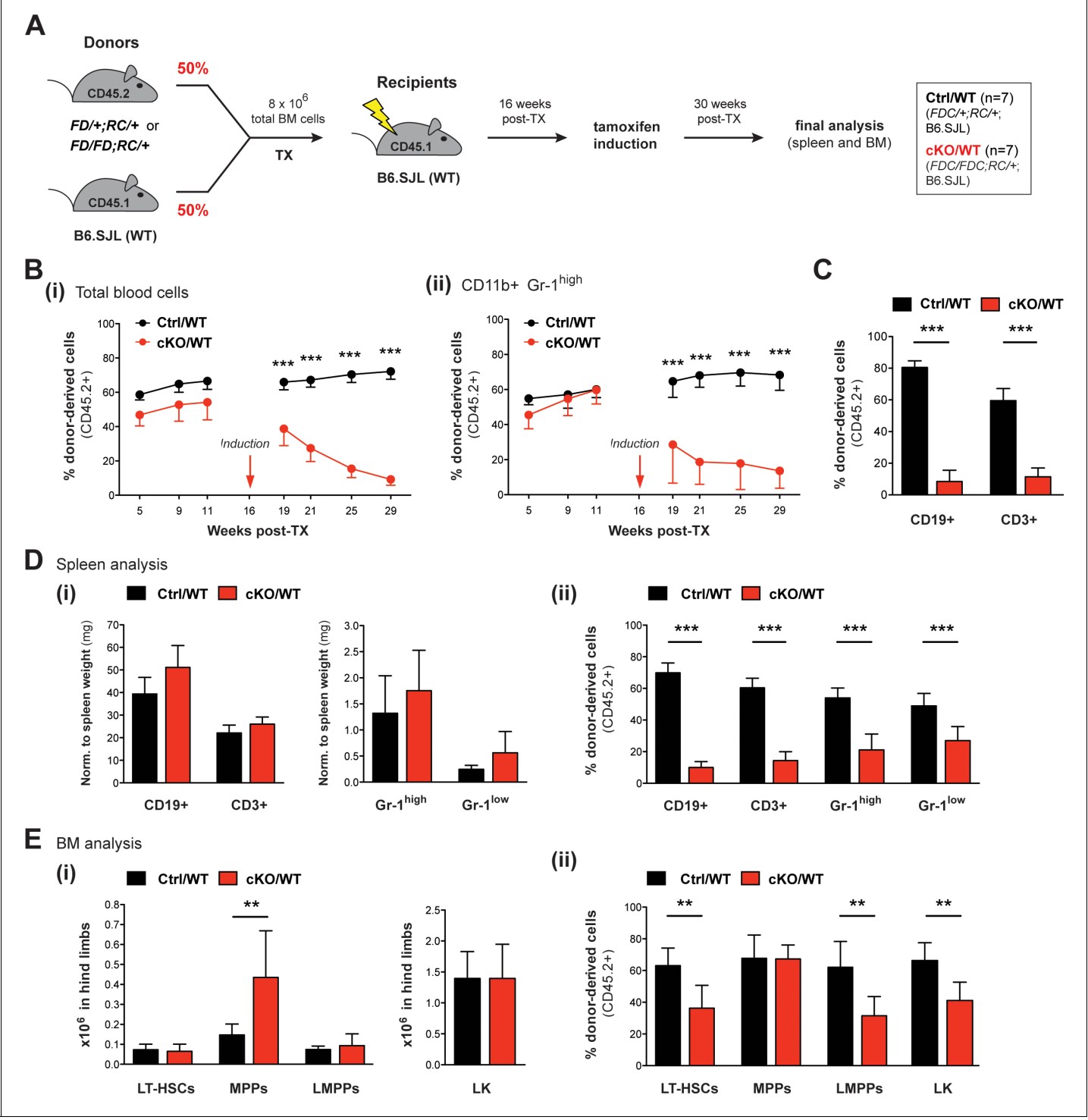

**Figure 8.** Competitive transplantation experiments reveal crucial requirements of Setd1b in HSPCs as well as myeloid and lymphoid lineages. (**A**) Transplantation (TX) setup. BM cells from tamoxifen-inducible heterozygous *Setd1b^FD/+;RC/+* or homozygous *Setd1b^FD/FD;RC/+* donor mice (CD45.2+) were intravenously injected together with WT cells (1:1) into lethally irradiated B6.SJL recipients (CD45.1+) (8 × 10^6 cells per TX). Tamoxifen was applied 16 weeks after TX and the mice (Ctrl/WT n = 7, cKO/WT n = 7) were analyzed 14 weeks later. (**B**) FACS analysis of the CD45.1/2 chimerism in peripheral blood over time. (i) Immediately after tamoxifen induction, the contribution of CD45.2+ donor-derived cells severely declined in cKO/WT transplanted mice (p***<0.001). (ii) In CD11b+ Gr-1^high granulocytes, the CD45.2+ ratio similarly dropped to < 10% (p***<0.001). (**C**) FACS analysis of the CD45.1/2 chimerism in peripheral blood lymphocytes 13 weeks after induction. In both CD19+ B cell and CD3+ T cell populations, the contribution of CD45.2 + donor-derived cells was significantly reduced in cKO/WT transplanted mice (p***=0.001). (**D**) FACS analysis of splenic cell compartments (Ctrl/WT

*Figure 8 continued on next page*

*Figure 8 continued*

n = 7, cKO/WT n = 6) 14 weeks after induction. (**i**) Absolute cell numbers (with respect to spleen weight) were comparable between cKO/WT transplanted mice and controls. (**ii**) FACS analysis of the CD45.1/2 chimerism in the respective cell compartments. In cKO/WT transplanted mice, the contribution of CD45.2+ donor-derived cells significantly decreased in all analyzed populations (p***=0.001). (**E**) FACS analysis of HSPCs in BM (Ctrl/WT n = 7, cKO/WT n = 6) 14 weeks after induction. (**i**) No major changes in overall HSPC numbers detected in cKO/WT transplanted mice, except for slightly increased MPPs (p**=0.003). (**ii**) FACS analysis of the CD45.1/2 chimerism in the respective cell compartments. Except for MPPs, the contribution of CD45.2+ donor-derived cells was significantly reduced in LT-HSCs (p**=0.005), LMPPs (p**=0.008), and LK cells (p**=0.008) in BM from cKO/WT transplanted mice. All graphs depict the mean ± SD.

DOI: https://doi.org/10.7554/eLife.27157.020

The following source data is available for figure 8:

**Source data 1.** Numerical data used to generate *Figure 8E* (LSK BM competitive TX).

DOI: https://doi.org/10.7554/eLife.27157.021

MPPs were the only cell type, in which the contribution of CD45.2+ cells was unchanged and comparable to controls, indicating that Setd1b was largely dispensable for their function (*Figure 8E–ii*). However, all other HSPC populations revealed a significant reduction in their CD45.2+ donor-derived cell compartment (*Figure 8E–ii*). Since in a competitive setting with WT BM, normal hematopoiesis rescues any deficits caused by Setd1b deletion, cKO HSPCs were relieved of the necessity to maintain homeostasis. This situation allowed unmasking cell-autonomous effects of Setd1b deletion, which we found to encompass both HSCs as well as myeloid and lymphoid progenitors with the notable exception of MPPs.

## Setd1b serves as a crucial regulator of normal proliferation and differentiation potential in HSPCs

To further identify initial changes upon loss of Setd1b, we performed mRNA expression profiling using the LSK compartment of control ($Setd1b^{FDC/+;RC/+}$) and cKO ($Setd1b^{FDC/FDC;RC/+}$) littermates (n = 3 each) 3 days after the last tamoxifen gavage. At this early stage, the cellular composition of the LSK compartment in cKO mice was largely comparable to controls (data not shown). The obtained reads showed good correlation among the biological replicates (*Figure 9A*). Selected mRNAs were validated by qRT-PCR with good overall agreement (*Figure 9—figure supplement 1A*). Although similar numbers of differentially expressed genes (DEGs) were up- (635 DEGs) and downregulated (671 DEGs) at a false discovery rate (FDR) of 5%, twice as many genes were downregulated than upregulated when considering changes of >2-fold (*Figure 9B*; *Figure 9—source data 1*). This accords with expectations for an H3K4 methyltransferase to sustain gene expression. The most highly enriched pathways in the group of downregulated DEGs related to mitochondrial and metabolic processes as well as protein degradation pathways (*Figure 9C*; *Figure 9—source data 2*). This suggests that cKO cells have reduced metabolic activity, which potentially explains the proliferative deficit seen *in vitro* (*Figure 6—figure supplement 2B-ii*). On the other hand, the top upregulated terms included signaling pathways involved in regulation of stem cell quiescence, differentiation, and inflammatory signaling (e.g. IL-8, NF-kB), which may indicate early compensatory responses to stress (*Figure 9C*; *Figure 9—source data 3*). This assumption gains further support from enriched disease terms like 'organismal injury' and 'inflammatory response' in the upregulated DEGs (*Figure 9—figure supplement 1B*). Alternatively, some of the upregulated DEGs (e.g. *c-Mpl*, *Tek*) may be a consequence of downregulation of Setd1b-dependent repressors as we observed for Setd1b in oogenesis (*Brici et al., 2017*). In accordance with this notion, several transcription factors (e.g. *Cebpa*, *Gata1*, *Klf1*) and differentiation-specific genes (e.g. *Mpo*, *Elane*, *Pf4*, *Ighm*, *Il7r*, *Dntt*) involved in the hematopoietic lineages were amongst the most highly downregulated genes (*Figure 9D*). To validate that the observed expression changes were not biased by an altered LSK composition in cKO BM, we performed qRT-PCR analysis of selected mRNAs in FACS-sorted HSPC populations from two representative mouse pairs (*Figure 9—figure supplement 1C*). The expression levels in cKO mice of *Klf1*, *Mpo,* and *Ighm* were consistently downregulated in LT-HSCs, MPPs, and LMPPs, respectively. Altogether, these findings strengthen the hypothesis that Setd1b affects expression of key hematopoietic genes that are important for cell fate choice and lineage specification.

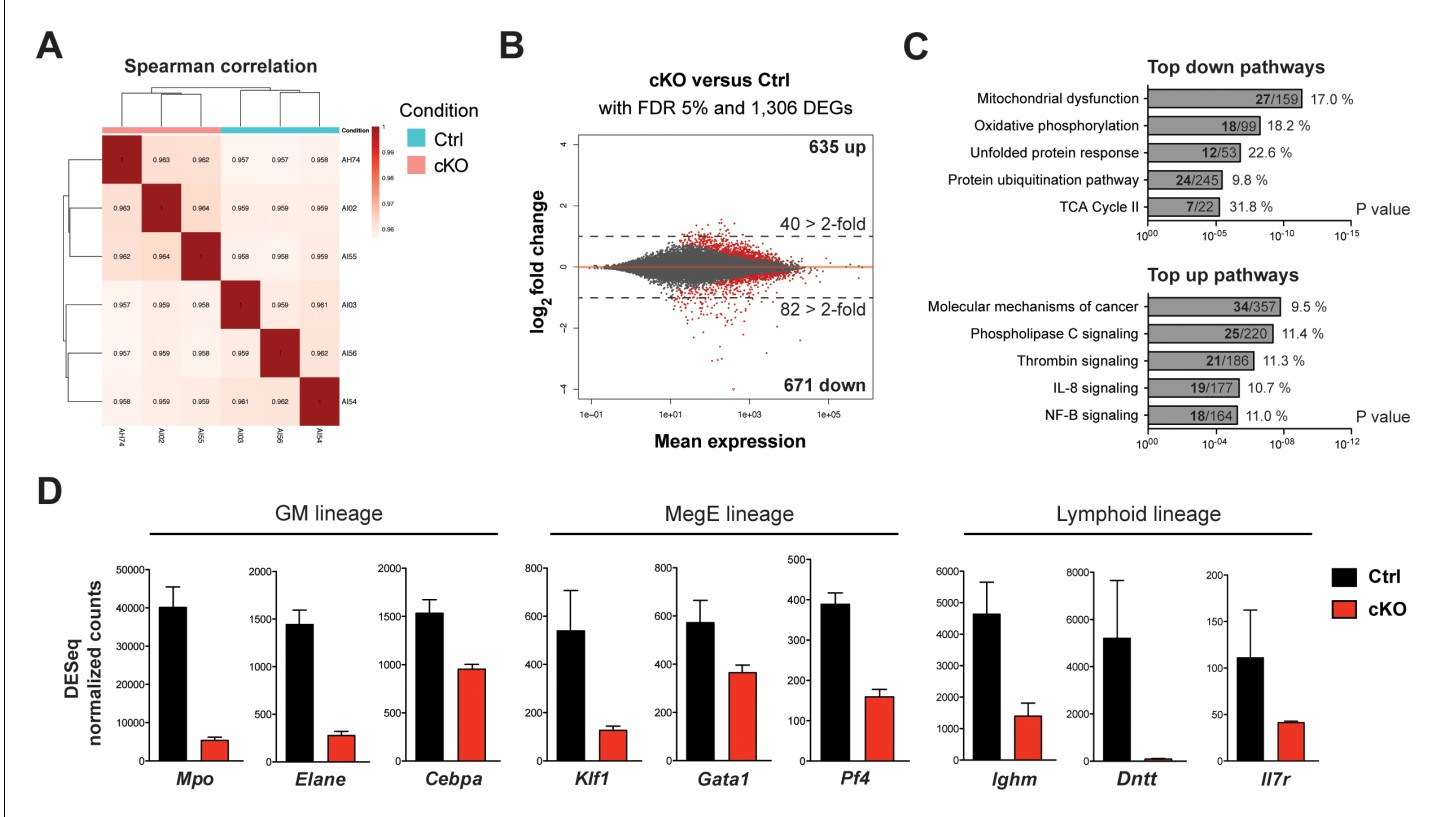

**Figure 9.** mRNA expression profiling implicates Setd1b as an essential factor for normal proliferation and differentiation of HSPCs. To reflect the initial changes upon loss of Setd1b, the LSK compartment from control ($Setd1b^{FDC/+;RC/+}$) and cKO ($Setd1b^{FDC/FDC;RC/+}$) littermates (n = 3) was isolated 3 days after the tamoxifen induction was completed. mRNA expression profiling yielded on average 35 million reads per sample. (**A**) Spearman correlation. All biological replicates mapped well together, indicating good reproducibility and data quality. (**B**) Nearly equal amounts of up- (635) and downregulated (671) differentially expressed genes (DEGs) were detected applying a false discovery rate (FDR) of 5%. However, twice as many DEGs were > 2-fold downregulated (82) than upregulated (40) as indicated by the dashed line at $\log_2 = 1$. All red data points refer to significant changes (adjusted p value < 0.05). (**C**) Pathway analysis using IPA software. In the group of downregulated DEGs, metabolic processes and protein degradation pathways were amongst the most enriched terms, suggesting that cKO cells were metabolically less active. In the category of upregulated DEGs, several signaling pathways relating to cancer and inflammation were enriched. For each pathway the total number of affected DEGs (bold) and percentages are given with respect to total number of genes within one group. (**D**) DESeq normalized counts of selected genes. Some of the most significantly downregulated genes (>1.6-fold) could be ascribed to transcription factors important for lineage specification and differentiation-specific genes. Each graphs depicts the mean ± SD. (GM = granulocyte/monocyte, MegE = megakaryocyte/erythroid).
DOI: https://doi.org/10.7554/eLife.27157.022

The following source data and figure supplement are available for figure 9:

**Source data 1.** Data used to generate *Figure 9B* (DEGs > 2-fold).
DOI: https://doi.org/10.7554/eLife.27157.024
**Source data 2.** Data used to generate *Figure 9C* (downregulated DEGs).
DOI: https://doi.org/10.7554/eLife.27157.025
**Source data 3.** Data used to generate *Figure 9C* (upregulated DEGs).
DOI: https://doi.org/10.7554/eLife.27157.026
**Figure supplement 1.** Validation of selected mRNAs by qRT-PCR and additional IPA analysis.
DOI: https://doi.org/10.7554/eLife.27157.023

## Discussion

The hematopoietic system is the original stem cell paradigm. Through intensive examination, the roles and hierarchies of transcription factors in hematopoietic stem cells and their differentiation programs are well documented (*Graf, 2002*; *Bodine, 2017*). Accumulating evidence implies that epigenetic regulation through the Set1/Trithorax system also contributes to the hierarchies of hematopoietic regulation (*Santos et al., 2014*; *Yang and Ernst, 2017*; *Arndt et al., 2018*;

*Beerman, 2018*). The first mammalian *Trithorax* homolog, *MLL1*, was originally discovered because it is commonly mutated in human leukemia (*Ziemin-van der Poel et al., 1991*; *Li and Ernst, 2014*). It was later realized that yeast Set1, *Drosophila* Trithorax, and Mll1 are H3K4 methyltransferases and consequently implicated in epigenetic regulation (*Roguev et al., 2001*; *Milne et al., 2002*). Mammals have six Set1/Trithorax-type H3K4 methyltransferases: Mll1 - Mll4, Setd1a, and Setd1b (*Glaser et al., 2006*). The functional studies reported here show for the first time a vital and distinct role for Setd1b in hematopoiesis. Our data, together with the thorough dissection of Mll1 (*Ernst et al., 2004*; *Jude et al., 2007*; *Chen et al., 2017*) and emerging information about Setd1a, Mll2, Mll3, and Mll4 (*Santos et al., 2014*; *Yang and Ernst, 2017*; *Arndt et al., 2018*) indicate that the contribution of Set1/Trithorax-type epigenetic regulators to transcriptional activity presents a second framework in addition to the transcription factor hierarchy for hematopoietic specification.

Initial identification of the Setd1b contribution to hematopoiesis emerged from ubiquitous *Rosa26-Cre-ERT2* (*Seibler et al., 2003*; *Glaser et al., 2009*) ligand (tamoxifen)-induced site specific recombination (*Logie and Stewart, 1995*). All Setd1b conditionally ablated (cKO) adult mice showed severe hematopoietic defects and died. To focus on the hematopoietic role of Setd1b, hematopoietic-specific deletion by *Vav-Cre* (vKO) (*Stadtfeld and Graf, 2005*) produced a fully penetrant lethal phenotype in the adult. Transplantation of Setd1b conditional HSC-enriched bone marrow (BM) into lethally irradiated WT recipients followed by tamoxifen induction as well as transplantation of vKO BM recapitulated the Setd1b-deficient phenotype, thereby proving an intrinsic requirement of Setd1b in the hematopoietic system. In these studies, one discrepancy arose. Most mice from the cKO and vKO BM transplantation experiments did not die prematurely as expected. Two explanations are plausible. Either Setd1b contributions from the non-transplanted stroma are the source of the difference or both cKO and vKO transplantation models included residual HSCs that were not abolished by the irradiation procedure and were able to partly restore hematopoietic activity and ensure survival. Careful examination of the contribution of CD45.2 + donor-derived cells in different hematopoietic compartments revealed severe decline in almost every cell type suggesting that the second explanation is more likely.

Irrespective of the KO strategy, we observed several common hallmark features associated with Setd1b-deficient hematopoiesis, including peripheral thrombo- and lymphocytopenia, multilineage dysplasia, and varying degrees of splenomegaly caused by myeloid-biased extramedullary hematopoiesis (*Table 1*). In addition, analysis of the HSPC compartment in BM reproducibly showed a severe shortage of lymphoid-primed multipotential progenitors (LMPPs), potentially reflecting the loss of both B and T cells in the periphery (*Table 1*). Concurrently, as predominantly noted in cKO mice, enriched numbers of MPPs possibly relate to the overproduction of myeloid cells and megakaryocytes in the BM. Competitive transplantation experiments permitted us to distinguish between the cell-autonomous effects of Setd1b deletion from those that were superimposed due to compensatory mechanisms. In the absence of peripheral cytopenia, the contribution of CD45.2+ cKO cells declined in LT-HSCs as well as hematopoietic progenitors in mixed cKO/WT BM chimeras. Notably, in the MPP compartment the contribution of cKO cells neither decreased nor increased. Altogether, this strongly suggests that (1) Setd1b plays an essential role in both hematopoietic stem and progenitor cells, (2) Setd1b function is indispensable for differentiation of both lymphoid and myeloid lineages, and (3) the consequential loss of mature blood cells elicits compensatory stress-induced hematopoiesis in the non-competitive setting (*Zhao et al., 2014*; *Zhao and Baltimore, 2015*). The resulting stimulation and peripheral mobilization of Setd1b-deficient HSPCs leads to enhanced myeloid-biased differentiation. The limited myeloproliferative phenotype in vKO mice could be explained by the gradual loss of Setd1b during fetal development. By affecting a still naïve hematopoietic system, the loss of Setd1b might result in an overall reduced capacity to adjust cellular output in response to stress. Eventually, this chronic insufficiency leads to stem cell exhaustion and increased propensity for clonal evolution, which accords with the development of MPN in some aging vKO mice.

Except for one case study of polycythemia vera (*Tiziana Storlazzi et al., 2014*), no *SETD1B* mutations in human hematologic malignancies such as myeloproliferative neoplasm (MPN), myelodysplastic syndrome (MDS), and AML have been reported so far. However, major phenotypic hallmarks of the Setd1b vKO mouse (e.g. multilineage cytopenia and dysplasia) are reminiscent of human MDS according to WHO classification and adapted diagnostic criteria in mice (*Kogan et al., 2002*; *Vardiman et al., 2009*; *Zhou et al., 2015*). Especially considering the need for suitable mouse

**Table 1.** Hematopoietic parameters of Setd1b-deficient mouse models

A summary table of (abnormal) hematopoietic parameters in Setd1b-deficient mouse models (cKO and cKO transplant; vKO and vKO transplant) is shown. All depicted changes are based on comparisons with heterozygous controls. (n.d. = not determined).

| | Phenotype | cKO | cKO transplant | vKO | vKO transplant |
|---|---|---|---|---|---|
| BM | LT-HSCs | Unchanged | Unchanged | Unchanged | Low |
| | MPPs | High | High | Unchanged | High |
| | LMPPs | Low | Low | Low | Low |
| | LK cells | Unchanged | Unchanged | Low (GMPs) | Unchanged |
| Spleen | B cells | Low | Low | Low | Low |
| | T cells | Unchanged | Unchanged | Low | Unchanged |
| | Granulocytes | High | High | High | High |
| | Extramedullary hematopoiesis | Yes | n.d. | n.d. | n.d. |
| | Splenomegaly | High | Low | Low | no |
| Blood | Lymphocytes | Low | Low | Low | Low |
| | Neutrophils | High | Unchanged | Unchanged | Unchanged |
| | Thrombocytes | Low | Low | Low | Low |
| | Red blood cells | Low | Unchanged | Low | Unchanged |
| | Dysplastic signs | Yes | Yes | n.d. | n.d. |
| | Immature myeloid forms | Yes | Yes | Yes | n.d. |

DOI: https://doi.org/10.7554/eLife.27157.027

models that recapitulate human disease parameters, we suggest that further investigation of the Setd1b-deficient phenotype will shed light on the molecular events driving myeloid neoplasia in the context of epigenetic dysregulation.

To illuminate the processes and gene expression programs initially altered upon loss of Setd1b, we performed mRNA expression profiling using the LSK compartment shortly after ablation of Setd1b. The most enriched pathways in the category of downregulated genes related to general cell metabolism, including mitochondrial energy production and protein degradation pathways, indicating a state of reduced proliferative activity in Setd1b-deficient HSPCs. This is concordant with the failure of Setd1b-deficient c-Kit+ cells to be immortalized *in vitro* by the strong *MLL-ENL* oncogene. Among the most highly downregulated genes we found transcription factors important for lineage specification (e.g. *Cebpa*, *Gata1*, *Klf1*) as well as lineage-specific genes (e.g. *Mpo*, *Elane*, *Pf4*, *Ighm*, *Il7r*, *Dntt*). In this capacity, Setd1b is apparently sustaining the expression of key lineage determination factors whose loss may not only provoke lineage abnormality but also impair or delay differentiation. Interestingly, many of the affected genes are co-expressed in HSPCs at low levels and become selectively upregulated during lineage commitment; a process called multilineage priming based on poised activation of lineage-specific enhancers (*Månsson et al., 2007*; *Mercer et al., 2011*; *Nimmo et al., 2015*). Whether Setd1b directly binds to enhancer elements or promoters of these genes or influences their expression via non-catalytic functions, remains subject to the development of an effective Setd1b chromatin immunoprecipitation approach suitable for limited cellular inputs.

All six mammalian Set1/Trithorax proteins individually reside in large complexes with an identical 'WRAD' protein scaffold (*Dou et al., 2006*; *Ruthenburg et al., 2007*) complemented by specific subunits. Rbm15 (Ott1), a pleiotropic regulator of transcription, mRNA splicing and export, interacts with Setd1b via its SPOC domain and appears to be a Setd1b-specific complex subunit (*Zolotukhin et al., 2009*; *Lee and Skalnik, 2012*; *Xiao et al., 2015*). Similar to the Setd1b-deficient phenotype, deletion of Rbm15 in adult mice causes myeloid and megakaryocyte expansion in spleen

and BM, impaired B cell differentiation, and an increased stem cell compartment (*Raffel et al., 2007*; *Niu et al., 2009*). This suggests that both proteins cooperate to maintain hematopoietic homeostasis, especially during proliferative stress (*Xiao et al., 2012*). Essential functions in the hematopoietic system have also been reported for Cfp1, which is a member of both Setd1a and Setd1b complexes (*Lee and Skalnik, 2005*). Similar to the Setd1b KO, Cfp1 ablation leads to rapid loss of lineage-specific progenitors and mature blood cells, while at the same time MPPs (LSK CD34 + Flt3-) accumulate (*Chun et al., 2014*). Likewise, conditional deletion of Dpy30 in the hematopoietic system causes defects of both self-renewal and differentiation in HSPCs (*Yang et al., 2016*). Although Dpy30 is a common subunit in all six H3K4 methyltransferase complexes, we suggest that Setd1b action contributes to the Dpy30-associated changes. Hematopoietic studies with the other Set1/Trithorax-type family members have revealed markedly different phenotypes, indicating a high level of specificity. Loss of Mll1 disturbs the transcriptional program required for HSC self-renewal and results in rapid BM failure (*Jude et al., 2007*; *Artinger et al., 2013*). Deletion of Mll4 using *Mx-Cre* resulted in increased numbers of LT-HSCs, myeloproliferation, and reduced lymphopoiesis (*Santos et al., 2014*). Setd1a, which is structurally closest to Setd1b, has a different function in hematopoiesis. In contrast to the adult lethal phenotype of Setd1b vKO mice, *Vav-Cre* mediated deletion of Setd1a revealed that definitive HSCs are not established and/or expanded during fetal development, leading to death before weaning (*Arndt et al., 2018*). In competitive transplantation experiments using the *Rosa26-Cre-ERT2* line, loss of Setd1b affected the LT-HSC population whereas loss of Setd1a had no effect on LT-HSCs (*Arndt et al., 2018*). Only after transplantation of BM cells from tamoxifen-induced Setd1a BM chimeras into secondary recipients, LT-HSCs were decreased. Thus, in agreement with the distinct requirements of Set1/Trithorax-type family members during embryonic development (*Yagi et al., 1998*; *Glaser et al., 2006*; *Lee et al., 2013*; *Bledau et al., 2014*) and oogenesis (*Yagi et al., 1998*; *Andreu-Vieyra et al., 2010*; *Brici et al., 2017*), each enzyme appears to fulfill a different vital role in hematopoiesis. Consequently, the analysis of Setd1b in blood adds to the emerging evidence that the six H3K4 methyltransferases sustain distinct aspects of cell-type-specific gene expression programs.

# Materials and methods

## Key resources table

| Reagent type (species) or resource | Designation | Source or reference | Identifiers | Additional information |
|---|---|---|---|---|
| Gene (*Mus musculus*) | SET domain containing 1B (Setd1b) | MGI:2652820 | | |
| Mouse strain | Setd1b[tm1.2Afst]/Setd1b+ | *Bledau et al., 2014*, PMID:24550110, *Brici et al., 2017*, PMID:28619824 | RRID:MGI:5570296 | |
| Mouse strain | C57BL/6-Gt(ROSA) 26Sort[m9(Cre/ESR1)Arte] | *Seibler et al., 2003*, PMID:12582257, Taconic | RRID:IMSR_TAC:10471 | |
| Mouse strain | B6.SJL-Ptprc[a] Pepc[b]/BoyCrl | Charles River | IMSR Cat# CRL:494, RRID:IMSR_CRL:494 | congenic |
| Antibody | anti-hSET1B | Bethyl | Bethyl Cat# A302-281A, RRID:AB_1850180 | 1:500 dilution |
| Antibody | anti-c-Kit | Thermo Fisher Scientific | Thermo Fisher Scientific Cat# 17-1171-82, RRID:AB_469430 | 1:400 dilution |
| Antibody | anti-Sca-1 | Thermo Fisher Scientific | Thermo Fisher Scientific Cat# 12-5981-82, RRID:AB_466086 | 1:800 dilution |
| Antibody | anti-CD34 FITC | Thermo Fisher Scientific | Thermo Fisher Scientific Cat# 11-0341-85, RRID:AB_465022 | 1:25 dilution |

*Continued on next page*

*Continued*

| Reagent type (species) or resource | Designation | Source or reference | Identifiers | Additional information |
|---|---|---|---|---|
| Antibody | anti-CD16/32 | Thermo Fisher Scientific | Thermo Fisher Scientific Cat# 25-0161-81, RRID:AB_469597 | 1:800 dilution |
| Antibody | anti-Flt3 | Thermo Fisher Scientific | Thermo Fisher Scientific Cat# 46-1351-82, RRID:AB_10733393 | 1:100 dilution |
| Antibody | anti-CD45.1 | Thermo Fisher Scientific | Thermo Fisher Scientific Cat# 25-0453-82, RRID:AB_469629 | 1:200 dilution |
| Antibody | anti-CD45.2 | Thermo Fisher Scientific | Thermo Fisher Scientific Cat# 11-0454-82, RRID:AB_465061 | 1:400 dilution |
| Chemical compound, drug | Tamoxifen | Sigma-Aldrich T5648 | | |
| Software, algorithm | BD FACSDiva Software | BD Biosciences | RRID:SCR_001456 | |

## Mice

The generation of the Setd1b conditional mouse line was previously described (*Bledau et al., 2014*). The targeted allele carries two selection cassettes, one upstream and one downstream of the loxP flanked frameshifting exon 5. Heterozygous mice were bred to the *CAGGs-Dre* line (*Anastassiadis et al., 2009*) to remove the downstream selection cassette, resulting in the constitutive 'D' allele. $Setd1b^{D/+}$ mice were then crossed to the *CAGGs-Flpo* line (*Kranz et al., 2010*) to remove the upstream selection cassette and to generate the conditional 'FD' allele. In this study, $Setd1b^{FD/+}$ mice were either crossed to the *Rosa26-Cre-ERT2* (RC) line (*Seibler et al., 2003*) to obtain tamoxifen-inducible $Setd1b^{FD/+;RC/+}$ mice or to the hematopoietic-specific *Vav-Cre* (VC) line (*Stadtfeld and Graf, 2005*) to obtain $Setd1b^{FD/+;VC/+}$ mice. All mouse strains were bred into a C57BL/6J genetic background. For tamoxifen inductions mice at an average age of 12 weeks received 5 doses of 4.5 mg tamoxifen (Sigma-Aldrich Chemie GmbH, Munich, Germany) via gavage (*Anastassiadis et al., 2010*). Complete recombination of the FD allele and generation of the frameshifted FDC allele was validated by PCR on genomic DNA. Primers for genotyping are provided in *Supplementary file 1A*. All animal experiments were performed according to German law and approved by the relevant authorities.

## Reverse transcription and qRT-PCR analysis

Total RNA from organs and cells was isolated using QIAzol Lysis Reagent (QIAGEN GmbH, Hilden, Germany). In case of FACS-sorted bone marrow (BM) cells, RNA was purified using the miRNeasy Micro Kit (QIAGEN) including on-column DNase I digestion. Reverse transcription of mRNA was achieved with the AffinityScript Multiple Temperature cDNA Synthesis Kit (Agilent Technologies, Santa Clara, CA). Quantitative PCR (qRT-PCR) was performed using GoTaq qPCR Master Mix (Promega GmbH, Mannheim, Germany) on a Mx3000P multiplex PCR instrument (Agilent Technologies). Ct values were normalized against *Rpl19* and fold changes in expression levels compared to controls were calculated according to the $2^{-\Delta\Delta CT}$ method (*Livak and Schmittgen, 2001*). Primer sequences and amplicon lengths are provided in *Supplementary file 1B*.

## Western blot analysis

Organs were disrupted in extraction buffer (20 mM HEPES, 350 mM NaCl, 2 mM EDTA pH 8.0, 10% glycerol, 0.1% Tween 20, 1 mM PMSF, 1x cOmplete Protease Inhibitor Cocktail (Sigma-Aldrich), 1 M DTT) using a POLYTRON homogenizer (KINEMATICA, Luzern, Switzerland) followed by three cycles of freezing and thawing. Extracts were incubated for 1 hr at 4°C with Benzonase Nuclease (125 U/ml; Merck Millipore, Darmstadt, Germany) to release nuclear proteins. 100 µg of whole cell protein extracts were separated using NuPAGE Novex™ 3–8% Tris-Acetate protein gels (Thermo Fisher Scientific, Schwerte, Germany) for 4 hr (80 V). After overnight transfer (8 V) to PVDF membranes, blots

were probed with primary antibodies in blocking solution (2% milk powder/1% BSA in 0.05% Tween 20/PBS): rabbit polyclonal anti-hSET1B (1:500; A302-281A BETHYL Laboratories, Montgomery, TX) and rabbit monoclonal anti-α-tubulin (1:2000; 1878–1 Epitomics, Burlingame, CA). Following incubation with the goat anti-rabbit secondary IgG HRP antibody (1:50,000; 31460 Thermo Fisher Scientific), chemiluminescence was detected on an ImageQuant LAS 3000 system (GE Healthcare, Munich, Germany).

## Blood count analysis and blood smears

Peripheral blood from the femoral vein was collected in EDTA-coated Microtainer Tubes (BD, Heidelberg, Germany). Blood counts were measured using the ProCyte Dx Hematology Analyzer (IDEXX Laboratories, Westbrook, Maine) and the Sysmex TX2000 blood analyzer (Sysmex, Kobe, Japan). Blood smears were stained with Wright-Giemsa according to standard protocols. Differential cell count analysis was obtained by the CellaVision DM 1200 (CellaVision AB, Lund, Sweden).

## Histology and immunohistochemistry

Organs were fixed in 4% PFA/PBS overnight and hind limb bones were decalcified in Osteosoft solution (Merck Millipore) for 4–5 days. Samples were dehydrated and paraffin-infiltrated by the STP 420 tissue processor (Thermo Fisher Scientific). 1 µm sections of paraffin-embedded tissue were stained with hematoxylin and eosin (H&E) according to standard protocols. Immunohistochemistry was performed as previously described (*Bledau et al., 2014*) using the following primary antibodies: rabbit polyclonal anti-myeloperoxidase (1:100; ab9535 Abcam, Cambridge, UK), rabbit polyclonal anti-von Willebrand factor (1:500; ab9378 Abcam). After counterstaining with hematoxylin, images were obtained at an Olympus BX61 upright microscope (Olympus Europa, Hamburg, Germany).

## Cytospin preparations

BM was flushed from hind limb bones in 5% FCS/PBS and filtered through a Sefar Nitex nylon mesh (Sefar AG, Heiden, Switzerland). Splenocyte and hepatocyte cell suspensions were obtained by grinding the tissue against frosted glass slides, filtering, and suspending in 5% FCS/PBS. In case of spleen and BM, red blood cell (RBC) lysis was performed using RBC Lysis Buffer (eBioscience, San Diego, CA). Cytospins ($5 \times 10^5$ cells per sample) were generated by the Cytospin™ 4 cytocentrifuge (Thermo Fisher Scientific). After drying and fixation in methanol, cytospins were stained with May-Grünwald-Giemsa according to standard protocols.

## Flow cytometry

BM and splenocyte cell suspensions were obtained as described above. Peripheral blood was collected in 10% heparin/PBS followed by RBC lysis. Unless CD16/32 (FcγIII/II receptor) was included into the antibody panel, samples were treated with Fc-block (1:100; eBioscience) for 15 min to reduce unspecific background staining. All staining panels, antibodies, and dilutions for fluorescence-activated cell sorting (FACS) are listed in *Supplementary file 2A*. A description of the hematopoietic stem and progenitor cell (HSPC) surface markers is provided in *Figure 4—figure supplement 1A*. All stainings were performed in 5% FCS/PBS for 45 min using $10^7$ cells per sample. If biotinylated antibodies were included, samples were additionally incubated for 20 min with the Qdot605 ITK Streptavidin Conjugate Kit (1:400; Thermo Fisher Scientific). To enrich for HSPCs, lineage-positive (Lin+) cells were depleted by staining with biotinylated lineage-specific antibodies followed by magnetic cell separation using the MagniSort system from eBioscience (SAV negative selection beads; 1:10 or 1:20). To exclude dead cells, samples were resuspended in SYTOX Blue Dead Cell Stain (1:20,000; Thermo Fisher Scientific). FACS analysis was performed on a LSR II cytometer (BD) and LSRFortessa (BD) using FACSDiva software (BD). FACS sorting was performed on a FACSAria III cell sorter (BD). An illustration of the HSPC gating strategy is provided in *Figure 4—figure supplement 1B*.

## BrdU proliferation assay

Mice were injected i.p. with 5-Bromo-2'-deoxyuridine (BrdU) (50 µg/g body weight; Sigma-Aldrich) and sacrificed 6 hr later. BM was dissected and stained for HSPCs as described above (*Supplementary file 2A*). After treatment with the Fixation/Permeabilization Solution Kit (BD)

samples were incubated for 1 hr at 37°C with DNase I (30 µg/$10^6$ cells; eBioscience). Detection of incorporated BrdU was achieved by a BrdU-specific antibody coupled to eFluor450 (1:20; 48-5071-42 eBioscience). FACS analysis was performed on a LSR II cytometer (BD).

## BM reconstitution experiments

Congenic B6.SJL recipient mice were subjected to whole body gamma irradiation using a $^{137}$Cs source (GSR D1 or Biobeam GM 2000). To minimize radiation-induced tissue damage, a split dose of 700 cGy and 400 cGy (4 hr between irradiations) was applied when transplanting plain RC/+ BM, and 2 × 500 cGy (5 hr between irradiations) when transplanting plain VC/+ BM and mixed BM. For plain RC/+ transplantation (TX), BM from 7 weeks old donor mice (Setd1b$^{FD/+;RC/+}$ or Setd1b$^{FD/FD;RC/+}$) was stained with the antibody panel outlined in *Supplementary file 2B*. After purifying the HSC-enriched Lin-/CD3- population, 3 × $10^4$ cells were injected intravenously into each recipient mouse (8 x Setd1b$^{FD/+;RC/+}$ and 14 x Setd1b$^{FD/FD;RC/+}$). For plain VC/+ TX, 8 × $10^6$ total BM cells from 12 weeks old donor mice (Setd1b$^{FD/+;VC/+}$ or Setd1b$^{FD/FD;VC/+}$) were transplanted via intravenous injection into each recipient mouse (n = 7 for each genotype). For competitive TX, total BM cells from 19 to 22 weeks old Setd1b$^{FD/FD;RC/+}$ or Setd1b$^{FD/+;RC/+}$ and WT (B6.SJL) donor mice were mixed in a 1:1 ratio and 8 × $10^6$ BM cells were transplanted via intravenous injection into each recipient mouse (n = 7 for each genotype). The relative contribution of donor and recipient cells to the hematopoietic compartment was regularly monitored by FACS analysis for CD45.1 (B6.SJL) and CD45.2 (C57BL/6) (*Supplementary file 2B*). When reaching a donor contribution of >90% in peripheral blood, RC/+ transplanted mice, in both the non-competitive and competitive setting, were induced by tamoxifen treatment as described above.

## Colony-forming unit and serial replating assays

For colony-forming unit (CFU) assays, $10^4$ cells of total BM suspensions were plated in duplicates in MethoCult GF M3434 semi-solid medium (STEMCELL Technologies GmbH, Cologne, Germany). The numbers of CFUs were determined after 9 days. For serial replating assays, the c-Kit+ population was isolated using MACS MicroBead technology (Miltenyi Biotec GmbH, Bergisch Gladbach, Germany). Immortalization by *MLL-ENL* using retroviral transduction was performed as previously described (*Lavau et al., 1997*). 2 × $10^4$ cells were replated in duplicates every 5 days in methylcellulose and stained with INT to assess colony formation.

## mRNA expression profiling

BM from cKO (Setd1b$^{FDC/FDC;RC/+}$) and control (Setd1b$^{FDC/+;RC/+}$) males (n = 3 each) was harvested 3 days after the tamoxifen induction was completed. Enrichment and staining of HSPCs was performed as described above (*Supplementary file 2A*). Sorting of the Lin- Sca-1+ c-Kit+ (LSK) population yielded on average 25 × $10^3$ cells. Cells were collected and stored at −80°C in RNAlater RNA Stabilization Reagent (QIAGEN) until RNA was harvested using the miRNeasy Micro Kit (QIAGEN). The eluted RNA was subsequently reverse-transcribed and amplified using the SMARTer Stranded Total RNA-seq Kit – Pico Input Mammalian (Takara Bio Europe, Saint-Germain-en-Laye, France). Resulting libraries were pooled in equimolar quantities for 75 bp single-read sequencing on a HiSeq 2500 system (Illumina, San Diego, CA), resulting in about 35 (30-53) million reads per sample. FastQC (Babraham Bioinformatics) and RNA-SeQC (v1.8.1) (*DeLuca et al., 2012*) were used to perform a basic quality control of the sequence data. Fragments were then aligned to the mouse genome (GRCm38/mm10) using GSNAP (2015-12-31) (*Wu and Watanabe, 2005*; *Wu and Nacu, 2010*). Uniquely aligned fragments and gene annotations from Ensembl (v81) served as input for featureCounts (v1.5.0) (*Liao et al., 2014*) to obtain a table with read counts per gene. To account for differences in library size, raw read counts were normalized with the R package DESeq2 (v1.10.1) (*Anders and Huber, 2010*). Spearman correlations were calculated with R and plotted with the R package pheatmap. Identification of differentially expressed genes (DEGs) was done with DESeq2 accepting a maximum of 5% false discovery rate (FDR) (adjusted P value < 0.05). Subsequent identification of enriched pathways was performed using Ingenuity Pathway Analysis (IPA) software (Qiagen).

## Statistics

For blood counts, the sample size was computed at the beginning of the study based on preliminary data (Wilcoxon-Mann-Whitney test with $\alpha = 0.05$ and $\beta < 0.2$). For analysis of hematopoietic cell populations in BM and spleen, sample sizes ranged between $n = 6$ and $n = 10$ to enable significant findings while taking into account biological variability. The number of biological replicates is indicated for each experiment and refers to the number of different mice used. Exceptions apply to the quantitative analysis of CFU numbers (*Figure 6—figure supplement 2A*), where each bar represents the mean of two technical replicates (duplicate cultures), and to the qRT-PCR analysis in FACS-sorted HSPCs from representative mice (*Figure 9—figure supplement 1C*), where each bar represents the mean of three technical replicates. In general, all data were included in the analysis to reflect normal biological variability. If technical errors occurred or mice revealed incomplete recombination, data points were excluded. GraphPad Prism software (v5.0a) was used to perform all basic statistical analysis. Data is presented as mean and error bars indicate standard deviation (SD) or standard error of the mean (SEM) as specified in each figure legend. Statistical significance was tested using the non-parametric Mann-Whitney U test to guarantee reliable testing for low sample sizes without having to meet critical distribution assumptions. For verification of mRNA expression profiles by qRT-PCR, the unpaired t test was used. In case of grouped analyses with two independent variables (genotype and time), two-way ANOVA was applied. Exact p values are reported for each graph with $p < 0.05$ indicating significant changes.

## Accession numbers

The mRNA expression profiling data in this paper have been deposited in NCBI Gene Expression Omnibus under accession number GSE97976.

## Acknowledgements

We thank the Biomedical Services of the Max Planck Institute of Molecular Cell Biology and Genetics Dresden for excellent care of our mouse colonies. We are grateful to Kristin Bernhardt, Stefanie Weidlich, Mandy Obst, and Juliane Bläsche for excellent technical assistance and other lab members for helpful discussions. We thank the core facilities of the Biotechnology Center for providing assistance with flow cytometry (Katja Schneider, Anne Gompf) and histology (Susanne Weiche) as well as the Institute for Clinical Chemistry and Laboratory Medicine (University Hospital Dresden, Germany) for generation of blood smears. The *Vav-Cre* mouse line was kindly provided by Thomas Graf (Centre for Genomic Regulation, Barcelona, Spain).

## Additional information

### Funding

| Funder | Grant reference number | Author |
|---|---|---|
| Deutsche Forschungsgemeinschaft | SPP1463/2 KR2154/4-1 | Andrea Kranz |
| Else Kröner-Fresenius-Stiftung | Stipend to Alpaslan Tasdogan | Alpaslan Tasdogan |
| Deutsche Forschungsgemeinschaft | SPP1463 SL27/7-2 | Robert Slany |
| Deutsche Forschungsgemeinschaft | SFB1074 project A2 | Hans Jörg Fehling |
| Deutsche Forschungsgemeinschaft | SPP1463/2 STE903/5-1 | Adrian Francis Stewart |
| Deutsche Forschungsgemeinschaft | SFB655 project B1 | Konstantinos Anastassiadis |
| Deutsche Forschungsgemeinschaft | SFB655 | Andreas Dahl |

Dresden International PhD program — Kerstin Schmidt

The funders had no role in study design, data collection and interpretation, or the decision to submit the work for publication.

## Author contributions

Kerstin Schmidt, Conceptualization, Validation, Investigation, Methodology, Writing—original draft, Writing—review and editing; Qinyu Zhang, Validation, Investigation, Methodology; Alpaslan Tasdogan, Conceptualization, Methodology, Writing—review and editing; Andreas Petzold, Andreas Dahl, Data curation, Methodology; Borros M Arneth, Investigation, Methodology; Robert Slany, Conceptualization, Resources, Investigation, Methodology, Writing—review and editing; Hans Jörg Fehling, Conceptualization, Resources, Writing—review and editing; Andrea Kranz, Conceptualization, Supervision, Funding acquisition, Validation, Investigation, Methodology, Writing—review and editing; Adrian Francis Stewart, Konstantinos Anastassiadis, Conceptualization, Supervision, Funding acquisition, Writing—original draft, Writing—review and editing

## Author ORCIDs

Kerstin Schmidt http://orcid.org/0000-0002-9596-4026
Borros M Arneth http://orcid.org/0000-0002-9793-0970
Robert Slany http://orcid.org/0000-0002-2028-9759
Andrea Kranz https://orcid.org/0000-0002-7481-0220
Adrian Francis Stewart http://orcid.org/0000-0002-4754-1707
Konstantinos Anastassiadis http://orcid.org/0000-0002-9814-0559

## Ethics

Animal experimentation: All animal experiments were performed according to German law and approved by the relevant authorities (Permit numbers: TVA 1188; AZ 55.2-2532-2-485; TVV 41/2016).

## Decision letter and Author response

Decision letter https://doi.org/10.7554/eLife.27157.036
Author response https://doi.org/10.7554/eLife.27157.037

# Additional files

## Supplementary files

• Supplementary file 1 Primers. (**A**) All primers used for genotyping are depicted. (**B**) All primers used for qRT-PCR are depicted. (se = sense, as = antisense, bp = base pairs)
DOI: https://doi.org/10.7554/eLife.27157.028

• Supplementary file 2 FACS antibodies and staining panels. (**A**) All antibodies and staining panels used for flow cytometry are depicted. (**B**) All antibodies and staining panels used for flow cytometry in transplanted mice are depicted.
DOI: https://doi.org/10.7554/eLife.27157.029

• Transparent reporting form
DOI: https://doi.org/10.7554/eLife.27157.030

## Data availability

Sequencing data have been deposited in GEO under accession code GSE97976

The following dataset was generated:

| Author(s) | Year | Dataset title | Dataset URL | Database, license, and accessibility information |
|---|---|---|---|---|
| Anastassiadis K, | 2018 | Expression profile of hematopoietic | https://www.ncbi.nlm. | Publicly available at |

| Schmidt K | stem and progenitor cells (HSPCs) after conditional deletion of the histone 3 lysine 4 (H3K4) methyltransferase Setd1b in mice | nih.gov/geo/query/acc. cgi?&acc=GSE97976 | the NCBI Gene Expression Omnibus (accession no: GSE97976). |
| --- | --- | --- | --- |

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
