## [Decision Letter]

Thank you for submitting your article "The H3K4 methyltransferase Setd1b is essential for hematopoietic stem and progenitor cell homeostasis in mice" for consideration by *eLife*. Your article has been favorably evaluated by Fiona Watt (Senior Editor) and three reviewers, one of whom, Amy J Wagers (Reviewer #1), is a member of our Board of Reviewing Editors. The following individual involved in review of your submission has agreed to reveal their identity: Ivan Maillard (Reviewer #3).

The reviewers have discussed the reviews with one another and the Reviewing Editor has drafted this decision to help you prepare a revised submission.

Summary:

Schmidt et al. investigate the function of Setd1b, an H3K4 methyltransferase, in the hematopoietic system using ubiquitous and blood-specific loss-of-function models and bone marrow transplantation. They show that loss of Setd1b causes loss of peripheral lymphocytes and platelets, enhanced extramedullary myelopoiesis, perturbed HSPC maintenance and premature mortality. These hematopoietic effects appear to be the major phenotype of adult-specific loss of Setd1b. They further implicate Setd1b in regulation of expression of *Cebpa, Gata1* and *Klf-1*. Altogether, the work is well executed and presented, providing new information to the field. Strengths include the novelty of the genetic model, as well as the interesting hematological phenotype. The results are potentially important because the possible redundant function of MLL/SET family members in vivo has not been well studied and because they strengthen the conclusion that Setd1b has an essential role in hematopoietic homeostasis in mice. Nonetheless, as outlined below, there are also several weaknesses in the experimental design and presentation that must be addressed to provide strong and compelling evidence for the authors' conclusions.

Essential revisions:

1) The biggest concern relates to the authors' conclusions regarding the role of the hematopoietic stem cell population in their Results, even though they used this term in the title of the manuscript. The authors state: "Using *Rosa26-Cre-ERT2* for near-ubiquitous ablation of Setd1b expression, we realized that the primary knockout phenotype in the adult is disturbed homeostasis of hematopoietic stem and progenitor cells (HSPCs) leading to hematopoietic failure and lethality." However, their data in Figure 4 and Figure 5 suggest that LT-HSCs are largely unaffected and the major changes are in the progenitor populations. The authors should provide some data supporting the concept that the HSC population is indeed affected by setd1b knock out, or they must rephrase their conclusions to be consistent with the data.

2) Similarly, they state: "Because we did not observe any other fatal organ failures or indicative symptoms, we suggest that the lethality of cKO mice was caused by hematopoietic deficiency." (presumably they mean hematopoietic deficiency of Setd1b). However, this conclusion is inconsistent with the results of the transplant studies, which do not show the same early lethality when Setd1b is deleted only in blood lineage cells. These data raise the distinct possibility that death of the Setd1b adult cKO mice is due to loss of Setd1b in another organ compartment, and this possibility should be discussed and the conclusions adjusted appropriately.

3) Figure 1, Figure 4 and Figure 6—figure supplement 1: The authors must confirm complete recombination in the blood system by PCR in different hematopoietic lineages, especially for hematopoietic stem cells, progenitor cells and proliferating myeloid cells. Incomplete recombination may confuse interpretation of these data.

4) Epigenetic analysis is not provided, and it is assumed but not proven that the effects are related to loss of SETD1B-mediated H3K4 methylation. This assumption may or may not be correct, given past findings with MLL1 in hematopoiesis (Ernst group), namely that the effects of MLL1 loss did not correlate with loss of methyltransferase activity in hematopoietic stem and progenitor cells.

5) Because *Vav1-Cre* can result in excision of loxP targets in some vasculature, as well as blood lineage cells, the authors should perform hematopoietic reconstitution experiments with the vKO model as well (similar to studies with the RosaCreER system). This would also address their suggestion (Discussion, second paragraph) that the less severe phenotype of the cKO transplanted mice relates to contributions from residual host cells.

6) Competitive transplantation experiments should be performed (as opposed to the non-competitive reconstitutions which the authors use in this paper) to better assess the cell-autonomous vs. non-autonomous effects of Setd1b in the absence of peripheral cytopenias. Well-performed competitive transplants could provide clarity and unify findings between the different model systems that have been reported. They could for example be performed with prior Setdb1 inactivation (to test HSC function in a competitive setting in parallel to reconstitution of individual lineages); and with Setdb1 inactivation after reestablishment of steady-state hematopoiesis (to test cell-autonomous effects of Setdb1 loss in individual lineages in a competitive setting that should rescue peripheral cytopenias).

We recognize that these particular experiments are time consuming and unless you have already initiated this work, it will likely take considerably longer than the two months we ordinarily allow for return of a revised manuscript. However, all agree this is essential and your work could not be reconsidered without this analysis.

7) Can the authors provide better clarity about the cause of death (hematopoietic failure? myeloproliferation? transformation?) in this model?

8) Transcriptomic comparison of LSK compartments of control and cKO mice (Figure 7) is problematic since LSK cells are heterogeneous and the authors show in Figure 4 that these the composition of the LSK subset is different in the two mouse genotypes (with cKO mice having an overrepresentation of MPPs and underrepresentation of LMPPs). The authors comment on this issue (subsection “Setd1b serves as a crucial regulator of normal proliferation and differentiation potential in HSPCs”), but it is important that they address it experimentally (e.g., by performing analysis, or at least validation by qPCR, in more homogeneous populations of LT-HSCs or other populations they determine are most affected by the KO).

---

## [Author Response]

Essential revisions:1) The biggest concern relates to the authors' conclusions regarding the role of the hematopoietic stem cell population in their Results, even though they used this term in the title of the manuscript. The authors state: "Using Rosa26-Cre-ERT2 for near-ubiquitous ablation of Setd1b expression, we realized that the primary knockout phenotype in the adult is disturbed homeostasis of hematopoietic stem and progenitor cells (HSPCs) leading to hematopoietic failure and lethality." However, their data in Figure 4 and Figure 5 suggest that LT-HSCs are largely unaffected and the major changes are in the progenitor populations. The authors should provide some data supporting the concept that the HSC population is indeed affected by setd1b knock out, or they must rephrase their conclusions to be consistent with the data.

The reviewers are concerned about the conclusion we made regarding the role of Setd1b in the hematopoietic stem cell compartment. The reviewers refer to the experiments presented in Figures 4 and 5, where we observed increased numbers of MPPs and reduction of LMPPs in the cKO mice but no changes in LT-HSCs. To prove that Setd1b was ablated in the LSK compartment we sorted the cells and tested by PCR for Cre-mediated recombination. We detected almost complete recombination in LT-HSCs, MPPs, LK cells and in granulocytes. Those results are presented in the new Figure 4—figure supplement 1. In the new transplantation experiments (competitive using the RC (*Rosa26-Cre-ERT2*) model and non-competitive using the VC (*Vav-Cre*) model) we detected a significant decrease of donor-derived Setd1b homozygous knockout LT-HSCs in both models as compared to controls. Those results are presented in the new Figures 7E-ii and 8E-ii and discussed in –the third paragraph of the Discussion. Therefore we believe that Setd1b plays a direct role in LT-HSCs in addition to maintaining homeostasis.

2) Similarly, they state: "Because we did not observe any other fatal organ failures or indicative symptoms, we suggest that the lethality of cKO mice was caused by hematopoietic deficiency." (presumably they mean hematopoietic deficiency of Setd1b). However, this conclusion is inconsistent with the results of the transplant studies, which do not show the same early lethality when Setd1b is deleted only in blood lineage cells. These data raise the distinct possibility that death of the Setd1b adult cKO mice is due to loss of Setd1b in another organ compartment, and this possibility should be discussed and the conclusions adjusted appropriately.

We removed the sentence to avoid early conclusions about the cause of lethality that were not sufficiently supported by evidence. At this time we could not exclude that in addition to hematopoietic defects other organs were also affected and contributed to lethality. However, with the hematopoietic specific deletion of Setd1b using the *Vav-Cre* model we observed a fully penetrant lethal phenotype (Figure 6B). We agree with the reviewers that there is a discrepancy regarding the lethality between Setd1b cKO mice and transplanted cKO mice (*Rosa-Cre-ERT2*). However, the recent non-competitive vKO and competitive cKO transplantation studies showed that donor-derived Setd1b-deficient HSPCs are diminished (new Figures 7E-ii and 8E-ii). This suggests that indeed host HSPCs, which were not abolished by the irradiation regime, are able to rescue hematopoiesis to an extent that ensures survival. We discuss this issue in –the second paragraph of the Discussion.

3) Figure 1, Figure 4 and Figure 6—figure supplement 1: The authors must confirm complete recombination in the blood system by PCR in different hematopoietic lineages, especially for hematopoietic stem cells, progenitor cells and proliferating myeloid cells. Incomplete recombination may confuse interpretation of these data.

Complete site-specific recombination is an important issue. We tested for recombination by PCR in the following FACS-sorted populations from the BM of cKO and vKO mice: LT-HSCs, MPPs, LK cells and CD11b+ Gr-1^high^ granulocytes. We detected complete recombination in all populations as defined by the absence of the unrecombined conditional allele in the PCR. These results are described in the first paragraph of the subsection “Hematopoietic stem and progenitor cell homeostasis is perturbed upon deletion of Setd1b” and –the first paragraph of the subsection “The hematopoietic-specific deletion of Setd1b by *Vav-Cre* is lethal and reveals functional deficits of HSPCs” and shown in the new Figure 4—figure supplement 2A-C and in the new Figure 6—figure supplement 1.

4) Epigenetic analysis is not provided, and it is assumed but not proven that the effects are related to loss of SETD1B-mediated H3K4 methylation. This assumption may or may not be correct, given past findings with MLL1 in hematopoiesis (Ernst group), namely that the effects of MLL1 loss did not correlate with loss of methyltransferase activity in hematopoietic stem and progenitor cells.

We did not test for changes in H3K4 methylation states upon loss of Setd1b. The main reason is that we do not have working protocols for assessing histone methylation in limited amount of material (low cell numbers). We did not intend to make the assumption that the effects are related to loss of Setd1b-mediated H3K4 methylation. We went through the text and did not find any sentences, which might lead to this assumption. From our data we cannot draw any conclusions whether Setd1b mediates its effects through catalytic or non-catalytic activity. Testing catalytically inactive methyltransferase domains as exemplified by P. Ernst with Mll1 are beyond the scope of this study. We discuss this as a future perspective in the fifth paragraph of the Discussion.

5) Because Vav1-Cre can result in excision of loxP targets in some vasculature, as well as blood lineage cells, the authors should perform hematopoietic reconstitution experiments with the vKO model as well (similar to studies with the RosaCreER system). This would also address their suggestion (Discussion, second paragraph) that the less severe phenotype of the cKO transplanted mice relates to contributions from residual host cells.

We agree with the reviewers that hematopoietic reconstitution experiments using the vKO model would be a valuable addition to the study. We performed non-competitive BM transplantations with *Setd1b FD/FD;Vav-Cre/+* and *FD/+;Vav-Cre/+* total BM cells into lethally irradiated B6.SJL recipients. In brief, vKO cells were able to reconstitute the BM niche. The results of the final analysis 16 weeks after transplantation largely mimic the vKO phenotype. Furthermore, analysis of the contribution of CD45.2+ donor-derived cells reveals the crucial function of Setd1b especially in LT-HSCs, LMPPs, and the lymphoid lineage. However, as we observed with the cKO transplantation model, the host cells seemed to rescue hematopoiesis to an extent that ensured survival of vKO transplanted mice up to the time of the final analysis. These results are described in the subsection “HSPCs, especially LT-HSCs and LMPPs, strongly depend on Setd1b function” and included as new Figure 7 and Figure 7—figure supplement 1.

Regarding Vav-Cre specificity, we used the line published by Stadtfeldt and Graf in 2005. This is a widely used Cre-deleter and according to literature excises only in hematopoietic cells (CD45+) and not in endothelial cells. Moreover, we analysed organs from *Setd1b FD/+;Vav-Cre/+* mice for recombination. We detected efficient recombination in hematopoietic organs (spleen and thymus) but not in organs with vascular networks of high density such as kidney and heart. Those results are shown in Author response image 1.

**Author response image 1. respfig1:** Efficient recombination occurs only in hematopoietic and not in non-hematopoietic organs of *Setd1b FD/+;Vav-Cre/+* mice. (**A**) PCR strategy to detect the WT, the unrecombined FD and the recombined FDC allele using primers upstream of the 5’ FRT site and downstream of the 3’ loxP site. (**B**) Efficient recombination as defined by the presence of the FDC band occurs only in thymus and spleen but not in heart and kidney.

6) Competitive transplantation experiments should be performed (as opposed to the non-competitive reconstitutions which the authors use in this paper) to better assess the cell-autonomous vs. non-autonomous effects of Setd1b in the absence of peripheral cytopenias. Well-performed competitive transplants could provide clarity and unify findings between the different model systems that have been reported. They could for example be performed with prior Setdb1 inactivation (to test HSC function in a competitive setting in parallel to reconstitution of individual lineages); and with Setdb1 inactivation after reestablishment of steady-state hematopoiesis (to test cell-autonomous effects of Setdb1 loss in individual lineages in a competitive setting that should rescue peripheral cytopenias).We recognize that these particular experiments are time consuming and unless you have already initiated this work, it will likely take considerably longer than the two months we ordinarily allow for return of a revised manuscript. However, all agree this is essential and your work could not be reconsidered without this analysis.

We agree with the reviewers that competitive transplantation experiments are essential for clarifying the role of Setd1b in different hematopoietic lineages and we thank them for giving us additional time for the revision in order to accomplish this. We decided to perform competitive transplantation experiments with Setd1b inactivation after re-establishment of steady-state hematopoiesis in order to ensure comparability with the previous non-competitive cKO transplantation model. Competitive BM transplantations were performed by injecting 50% *Setd1b FD/FD;RC/+* or *FD/+;RC/+* and 50% B6.SJL BM cells into B6.SJL recipients. Setd1b was inactivated by tamoxifen administration 16 weeks after transplantation. The results are presented in the new Figure 8 and described in the subsection “Competitive BM reconstitution reveals cell-autonomous effects of Setd1b in both myeloid and lymphoid lineages”. In brief, the results indicate that Setd1b is essential in both myeloid and lymphoid lineages and cell-autonomously required in LT-HSCs, LMPPs and early myeloid progenitors (LK compartment). The implications of these results with respect to the other Setd1b KO models are discussed in the third paragraph of the Discussion.

7) Can the authors provide better clarity about the cause of death (hematopoietic failure? myeloproliferation? transformation?) in this model?

This is difficult because we do not have experimental evidence that allows a clear distinction between hematopoietic failure and myeloproliferation as the cause of lethality. In Setd1b-deficient mice, impaired differentiation of HSPCs and the consequential loss of mature blood cells likely elicit compensatory myeloid-biased hematopoiesis, which further compromises hematopoietic function. We assume that this ultimately leads to hematopoietic failure. So far, transformation was only noted in the *Vav-Cre* model, however it was not fully penetrant.

8) Transcriptomic comparison of LSK compartments of control and cKO mice (Figure 7) is problematic since LSK cells are heterogeneous and the authors show in Figure 4 that these the composition of the LSK subset is different in the two mouse genotypes (with cKO mice having an overrepresentation of MPPs and underrepresentation of LMPPs). The authors comment on this issue (subsection “Setd1b serves as a crucial regulator of normal proliferation and differentiation potential in HSPCs”), but it is important that they address it experimentally (e.g., by performing analysis, or at least validation by qPCR, in more homogeneous populations of LT-HSCs or other populations they determine are most affected by the KO).

As we have stated in the manuscript, we aimed to measure early transcriptional changes caused by Setd1b deletion and therefore we performed mRNA expression profiling 3 days after the last tamoxifen application. At this early stage the cellular composition of the LSK compartment in cKO mice was largely comparable to controls (subsection “Setd1b serves as a crucial regulator of normal proliferation and differentiation potential in HSPCs”). Representative FACS plots from control and cKO BM that was used for mRNA expression profiling are shown in Author response image 2.

**Author response image 2. respfig2:** Representative FACS plots 3 days after the last tamoxifen application show that the cellular distribution in the LSK compartment was comparable between controls and cKO mice.

To validate our results, as suggested by the reviewers, we sorted LT-HSCs, MPPs and LMPPs from two mouse pairs (control and cKO) 3 days after the last tamoxifen gavage and performed qRT-PCR for 3 representative target genes (*Klf-1, Mpo and Ighm*). The results are included in updated Figure 9—figure supplement 1C and described in the subsection “Setd1b serves as a crucial regulator of normal proliferation and differentiation potential in HSPCs”.